# Gamma rhythms and visual information in mouse V1 specifically modulated by somatostatin[+] neurons in reticular thalamus

Mahmood S Hoseini[1†], Bryan Higashikubo[2†], Frances S Cho[2,3,4,5], Andrew H Chang[2,4], Alexandra Clemente-Perez[2,3,4,5], Irene Lew[2,4], Agnieszka Ciesielska[2,4], Michael P Stryker[1,3,5†], Jeanne T Paz[2,3,4,5†*]

[1]University of California, San Francisco, Department of Physiology, San Francisco, United States; [2]Gladstone Institute of Neurological Disease, San Francisco, United States; [3]University of California, San Francisco, Neurosciences Graduate Program, San Francisco, United States; [4]University of California, San Francisco, Department of Neurology, San Francisco, United States; [5]Kavli Institute for Fundamental Neuroscience, University of California San Francisco, San Francisco, United States

*For correspondence:
jeanne.paz@gladstone.ucsf.edu

†These authors contributed equally to this work

Competing interests: The authors declare that no competing interests exist.

**Abstract** Visual perception in natural environments depends on the ability to focus on salient stimuli while ignoring distractions. This kind of selective visual attention is associated with gamma activity in the visual cortex. While the nucleus reticularis thalami (nRT) has been implicated in selective attention, its role in modulating gamma activity in the visual cortex remains unknown. Here, we show that somatostatin- (SST) but not parvalbumin-expressing (PV) neurons in the visual sector of the nRT preferentially project to the dorsal lateral geniculate nucleus (dLGN), and modulate visual information transmission and gamma activity in primary visual cortex (V1). These findings pinpoint the SST neurons in nRT as powerful modulators of the visual information encoding accuracy in V1 and represent a novel circuit through which the nRT can influence representation of visual information.

## Introduction

Visual perception relies on the ability to focus on important information while ignoring distractions. Such selective attention is associated with neural oscillations in the gamma frequency band (~30–90 Hz, 'gamma oscillations') in the visual cortices in both rodents and humans (*Engel et al., 2001*; *Taylor et al., 2005*; *Pavlova et al., 2006*; *Doesburg et al., 2008*; *Ray et al., 2008*; *Siegel et al., 2008*). Gamma oscillations, particularly in the visual cortex in rodents, carnivores, and primates, are associated with a high level of cortical activity, and some have speculated that they may play a causal role in perception and the focusing of attention (*Gray et al., 1992*; *Singer and Gray, 1995*; *Kreiter and Singer, 1996*; *Yazdan-Shahmorad et al., 2013*) or in enabling a time-division multiplexing of cortical responses to multiple simultaneous stimuli (*Stryker, 1989*). Therefore, understanding the cellular and circuit mechanisms that underlie gamma rhythms could lead to a better understanding of the mechanisms behind higher cognitive functions such as perception and attention.

Two types of gamma oscillations have been reported in the primary visual cortex (V1), differentiated by whether they result from intra-cortical or subcortical inputs through the dorsal lateral geniculate nucleus (dLGN) in the mouse thalamus (*Saleem et al., 2017*). dLGN—the source of the specific sensory input to V1—gates the flow of all visual information to V1. The activity in dLGN is controlled by inputs from GABAergic neurons from the nucleus reticularis thalami (nRT) (*Houser et al., 1980*;

*Sherman and Koch, 1986*; *McCormick and Bal, 1997*; *Gentet and Ulrich, 2003*; *Lam and Sherman, 2011*; *Reinhold et al., 2015*; *Halassa and Acsády, 2016*; *Crabtree, 2018*; *Campbell et al., 2020*). Indeed, optogenetic activation of inhibitory (Gad2-positive) neurons in the dorsal portion of the nRT transiently reduces activity in the dLGN (*Reinhold et al., 2015*). Similar results were obtained with optogenetic activation of inhibitory neurons expressing somatostatin (SST) in transgenic mice (*Campbell et al., 2020*). However, it is unknown whether nRT can control gamma rhythms or the representation of visual information in V1, and if so, whether such control involves a specific cell type.

In our previous work we dissociated the connectivity, physiology, and circuit functions of neurons within rodent nRT, based on the expression of parvalbumin (PV) and SST markers, and validated the existence of such populations in human nRT (*Clemente-Perez et al., 2017*). Specifically, we showed in mice that (1) somatosensory nRT PV but not SST neurons exhibit intrinsic rhythmogenic properties due to the presence of low-threshold T-type calcium channels; (2) PV and SST neurons in the somatosensory nRT segregate into distinct input-output circuits; and (3) PV neurons are the main rhythm generators in the somatosensory circuits. However, it remained unknown whether PV and SST neurons also have distinct roles in other sensory sectors of the nRT.

A positive relationship between gamma power and the encoding of memories as assayed by retrieval has been widely reported for many years in the hippocampus and closely related cortical areas (*Wang, 2010* and references therein). In sensory cortex, however, it has not been clear whether gamma power is associated with the encoding of information (*Ray and Maunsell, 2015*). For this reason, we focused our study on the role of specific nRT cell types in thalamocortical visual processing on measurements of both field potentials and the fidelity of encoding of visual information in the firing activity of V1 neurons. Specifically, we investigated whether SST and PV nRT neurons modulate activity in the thalamocortical visual system. We began by examining the anatomy of cell-type-specific projections from nRT to the lateral geniculate nucleus (dLGN) and neighboring thalamic structures. Finding that the SST-cell projection was more prominent, we investigated their role in modulating responses in the thalamocortical visual system by optogenetically exciting or inhibiting them and making recordings in the V1 of the changes produced in local field potentials (LFPs) in different frequency bands and in the sensory responses of isolated single neurons. These sensory responses provide information about the visual world, and we computed changes in the representation of visual information from the single cellular responses of multiple neurons that were recorded simultaneously. The large effects of SST nRT cell activation on V1 were compared to the smaller and sometimes different effects of PV nRT cell activation. Finally, we studied the effects of SST nRT cell activation on the responses of single neurons in the dLGN, the site that receives direct input from nRT. Our electrophysiological findings, in line with anatomical data, indicate that activating SST but not PV neurons in nRT strongly reduces both visual information transmission and gamma power in V1 and dLGN.

## Results

### dLGN receives projections mainly from SST nRT neurons

To determine whether the visual thalamocortical relay nuclei receive inputs from SST and/or PV neurons of the nRT, we injected an AAV viral construct encoding enhanced yellow fluorescent protein (eYFP) in the nRT of adult *Sst-Cre* and *Pvalb-Cre* mice. We used the viral approach in adult mice because in transgenic knockin mice transient expression of PV and SST during early development causes labeling of the same cell population, precluding distinction between PV and SST cells in adulthood. We previously validated the viral approach immunohistochemically (*Clemente-Perez et al., 2017*). In adult mice, SST and PV neurons of the nRT were found to target distinct midline and somatosensory thalamocortical relay nuclei (*Clemente-Perez et al., 2017*). *Sst-Cre* and *Pvalb-Cre* cell bodies and their axons robustly expressed AAV-eYFP 4 weeks post-injection (*Figure 1A, B*), and we will refer to these as SST and PV neurons hereon. Confocal microscopy revealed dense axonal boutons from SST nRT neurons in the dLGN, but only very sparse input from PV nRT neurons (*Figure 1B–D*). This was surprising given that PV neurons are thought to represent the major cellular population of nRT (*Clemente-Perez et al., 2017*; *Li et al., 2020*). As previously shown, PV but not SST neurons from the nRT projected densely to the somatosensory

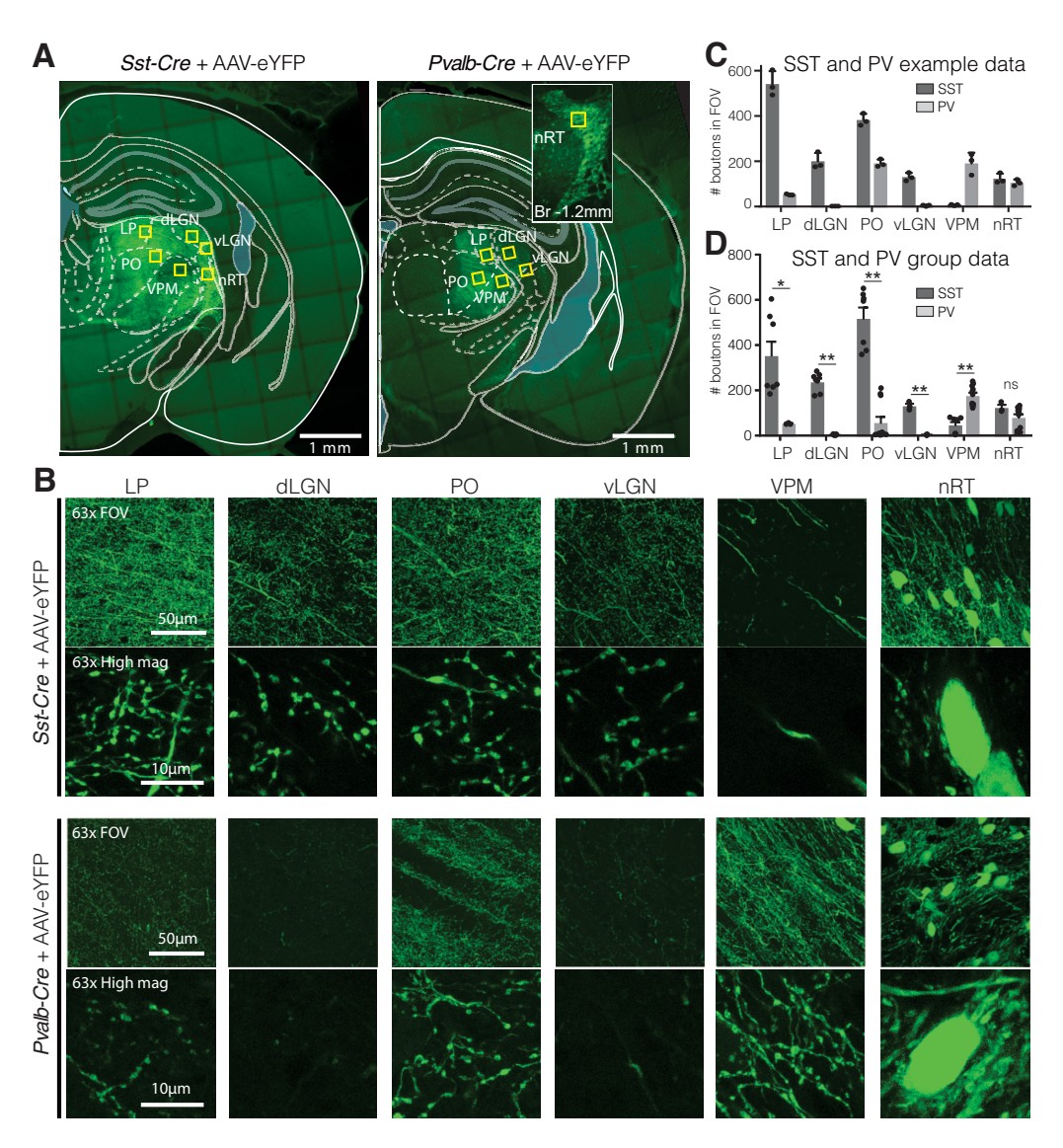

**Figure 1.** Visual relay thalamic nuclei are preferentially targeted by somatostatin (SST) and not parvalbumin (PV) GABAergic neurons from nucleus reticularis thalami (nRT). (A) Representative example sections of *Sst-Cre* and *Pvalb-Cre* mice after injection of floxed AAV in nRT, which results in enhanced yellow fluorescent protein (eYFP) expression in cell bodies and projections of SST or PV neurons, respectively. Yellow boxes indicate locations chosen for 63× confocal imaging and putative bouton quantification. Inset: nRT injection site as seen in an adjacent section. (B) 63× confocal images showing the entire field of view (FOV) and a zoomed cropped region ('High mag') to show details of axonal boutons and nRT somata. LP: lateral posterior nucleus; dLGN: dorsal lateral geniculate nucleus; PO: posterior medial nucleus; vLGN: ventral lateral geniculate nucleus; VPM: ventroposteromedial nucleus. The expression of the viral constructs in different brain regions was confirmed using the mouse brain atlas (*Paxinos and Franklin, 2001*). (C) Number of eYFP-labeled boutons present in thalamic nuclei of representative mice shown in (A). Data taken from three consecutive sections from each mouse. (D) Number of eYFP-labeled boutons present in thalamic nuclei of all mice imaged (n = 2 *Sst-Cre*, 3 *Pvalb-Cre*, 3–4 sections per mouse). Differences are significant between genotypes for all regions except for nRT after correction for multiple comparisons. *p<0.05, **p<0.01. The online version of this article includes the following figure supplement(s) for figure 1:

**Figure supplement 1.** Opsin expression in *Sst-Cre* and *Pvalb-Cre* mice.

**Figure supplement 2.** Retrograde labeling of nucleus reticularis thalami (nRT) neurons via intra-dorsal lateral geniculate nucleus (dLGN) injection of CTB-Alexa Fluor 488.

ventroposteromedial (VPM) thalamocortical nucleus (*Figure 1B–D*, *Figure 1—figure supplement 2*). Also, PV nRT neurons projected to the higher-order visual thalamus LP (*Figure 1B–D*, *Figure 1—figure supplement 2*), suggesting that lack of dense synaptic boutons in dLGN was not due to lack of

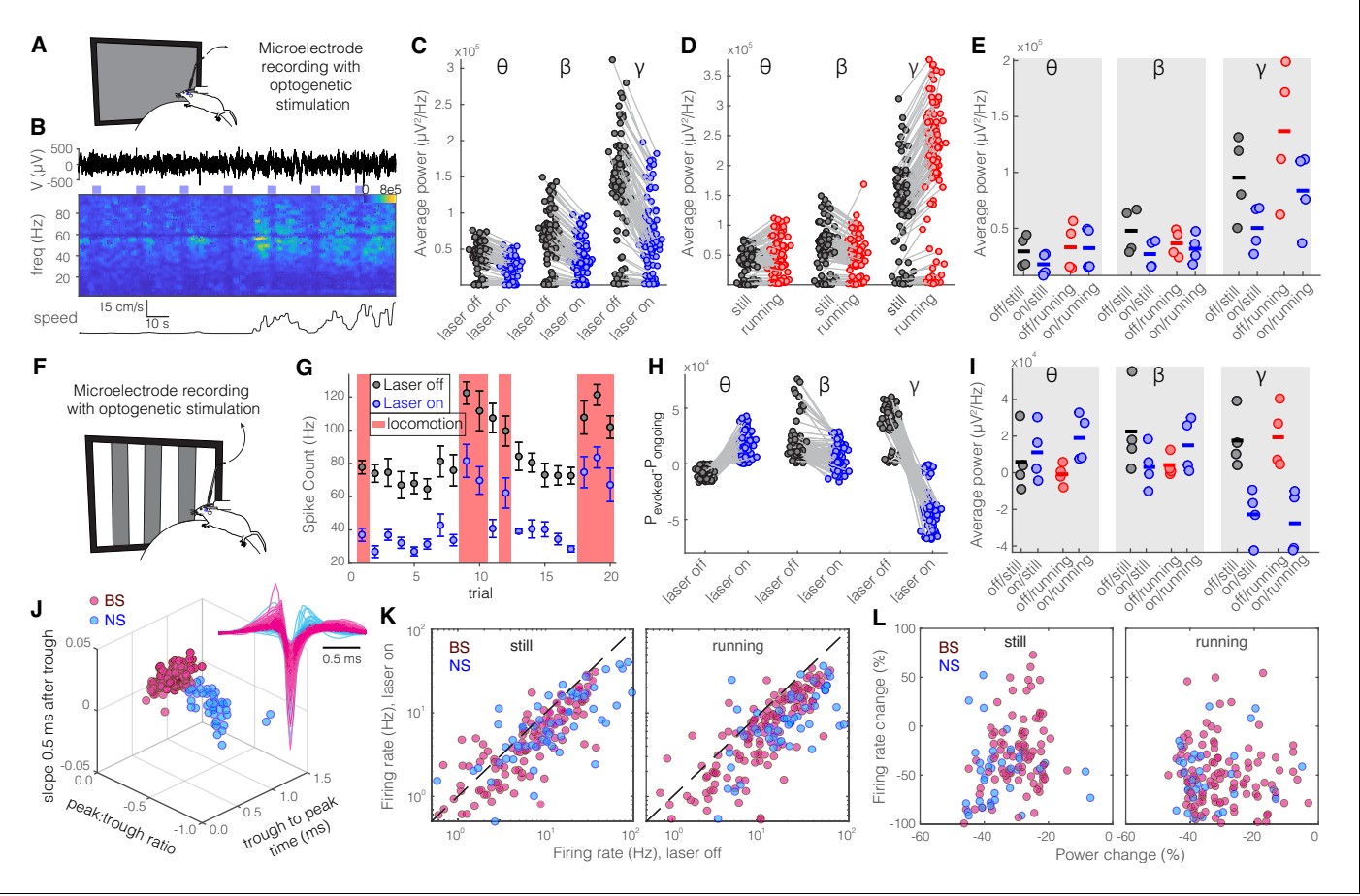

**Figure 2.** Optogenetic activation of somatostatin (SST) nucleus reticularis thalami (nRT) neurons reduces gamma activity in the primary visual cortex (V1) both with and without visual stimulation. (A) Neural activity was recorded from V1 in freely moving mice. Mice were presented with a gray blank screen while a blue light (473 nm, ~63 mW/mm$^2$) was delivered to Channelrhodopsin-2 (ChR2)-expressing SST cells in nRT using an optical fiber implanted above the nRT. Mouse movement was tracked over the course of the experiment. (B) Representative extracellular raw voltage trace is shown along with its power spectrum. Blue shading areas indicate optogenetic activation. Mouse movement speed is shown at the bottom. Note that optogenetic activation of SST cells in nRT reduced the power of local field potentials (LFPs) regardless of the locomotion state of the animal. (C) Optogenetic stimulations at baseline: across-trial average power of all channels in the absence (black) and presence (blue) of optogenetic activation in one representative mouse shows significant decreases across theta (4–8 Hz), beta (15–30 Hz), and gamma bands (30–80 Hz) (see *Table 1* for statistics). (D) Effect of locomotion without optogenetic stimulations: across-trial average power of all channels in one representative mouse indicates that locomotion has no significant effect on theta power, significantly reduces beta power, while causing a strong enhancement in gamma power (*Table 1*). (E) Effect of optogenetic stimulations in still and running conditions: average power across all channels in four mice shows that optogenetic activation of SST nRT cells selectively reduces gamma power both in the still (black vs. blue marks) and running conditions (red vs. blue marks) (*Table 1*), and has no significant effect on theta and beta powers from four mice. (F) Visual responses were recorded while mice were presented with moving gratings (eight directions, each moving in one of two possible directions; 2 s duration; randomly interleaved with optogenetic stimulation) in the visual field contralateral to the recording site. (G) Firing rate (averaged over all eight drifting directions) of an example cell during the course of the experiment. Black marks: visual responses when the laser is off; blue marks: visual responses when visual stimuli and optogenetic activation of SST nRT cells are coupled. Red shadings: locomotion state. Error bars: SEM. (H) Effect of visual stimulation with and without optogenetic manipulation: stimulus-evoked (average over all 20 trials) minus ongoing power of all channels when the laser is off (black circles) versus the laser-on condition (blue circles) in one representative mouse indicates a significant shift across all frequencies (*Table 1*). (I) Same as in (E) in the presence of visual stimulus (*Table 1*). Note that optogenetic activation of SST nRT cells selectively reduces gamma power in both still and running conditions, without significant effects on the other frequency bands. (J) Using three parameters calculated from average waveforms, cells were classified into narrow- (NS, cyan) or broad- (BS, magenta) spiking (height of the positive peak relative to the negative trough: −0.20 ± 0.01, −0.34 ± 0.02 [p=1.02e-9, Wilcoxon rank-sum test]; the time from the negative trough to the peak: 0.73 ± 0.02, 0.32 ± 0.02 ms [p=3.9e-33], slope of the waveform 0.5 ms after the negative trough: 0.01 ± 0.00, −0.01 ± 0.00 [p=5.94e-35], BS [n = 169], and NS [n = 73] cells, respectively). Subplot: average spike waveforms for all units, aligned to minimum, demonstrating BS (magenta) and NS (cyan) cells. (K) Optogenetic light significantly reduced firing rate of BS (magenta) and NS (cyan) cells during visual stimulus in both still and running conditions (*Table 1*). (L) Percentage change in visually evoked firing rate of both cell types versus percentage change of power in

*Figure 2 continued on next page*

*Figure 2 continued*

channels that each cell is recorded from for still and running states (*Table 1*, Spearman's rho and p: 0.05, 0.58 BS and still [n = 100 BS in four mice]; −0.12, 0.22 BS and running; 0.19, 0.29 NS and still [n = 32 NS in four mice]; 0.05, 0.77 NS and running).

The online version of this article includes the following source data and figure supplement(s) for figure 2:

**Source data 1.** Results of significance testing across different conditions.
**Figure supplement 1.** Visual stimuli consistently increase both power and average firing rate while visual exposure coupled with optogenetic activation causes a marked reduction in both measures.
**Figure supplement 2.** Optogenetic activation of somatostatin (SST) neurons in nucleus reticularis thalami (nRT) reduces visually evoked field responses in V1.
**Figure supplement 3.** Optogenetic activation of parvalbumin (PV) nucleus reticularis thalami (nRT) neurons produces an insignificant increase in gamma activity in V1 both with and without visual stimulation.

expression of the viral construct in PV nRT neurons. Furthermore, injections of the retrograde tracer cholera-toxin (CTB) in dLGN resulted in retrograde labeling of SST and PV neurons in the visual sector of the nRT (*Figure 1—figure supplement 1*). We cannot exclude that the retrograde labeling of certain nRT neurons might have resulted from tracer uptake by axons that are on route through the dLGN to LP. Nevertheless, these results suggest that SST and PV neurons projecting to first- and higher-order visual nuclei are intermingled in the same visual sector of the nRT. Retrograde labeling also revealed neurons that co-expressed SST and PV markers consistent with previous work (*Clemente-Perez et al., 2017*, *Figure 1—figure supplement 2*).

## Optogenetic activation of SST but not PV nRT neurons reduces gamma power in V1 both with and without visual stimulation

Given the distinct projections from SST and PV nRT neurons to dLGN, whose main target is V1 cortex, we next investigated to what extent perturbing the activity of SST and PV nRT neurons affects visual responses in V1. For this purpose, we injected an AAV viral construct containing Channelrhodopsin-2 (ChR2) in the nRT of *Sst-Cre* and *Pvalb-Cre* mice. The expression of opsins was restricted to the nRT, was cell-type-specific, and the opsins were well expressed throughout the visual sector of the nRT (*Sokhadze et al., 2019*; *Figure 1—figure supplements 1* and *2*). Given that we used saturating illumination to activate these neurons, the activation of SST-nRT input to dLGN was likely uniform rather than focal. Thereafter, extracellular recordings of single-unit activity and LFPs were made using a double-shank 128-channel microelectrode array placed in the V1 of mice that were free to stand or run on a polystyrene ball floating on an air stream (*Figure 2A*; *Du et al., 2011*; *Hoseini et al., 2019*). Mice viewed a gray blank screen while a blue light (473 nm, ~63 mW/mm$^2$) was delivered using an optical fiber implanted above nRT during different locomotion states (*Figure 2B*). Optogenetic activation reduced across-trial average power (scaled by 1/f) in all recording channels and all frequencies, but the strongest reduction was in the gamma band (*Figure 2C*, *Table 1*). Consistent with previous findings, locomotion itself differently modulated power across different frequencies (*Niell and Stryker, 2010*), and the main effect of locomotion was a dramatic power increase at higher frequencies in the gamma band (30–80 Hz) (*Figure 2D*, *Table 1*). Whether the mice were still or running, activation of SST nRT neurons significantly reduced gamma power, but not theta and beta power, in V1, compared to baseline (*Figure 2E*, *Table 1*).

To investigate how evoked visual responses are affected by optogenetic activation of SST nRT neurons, visual responses were recorded to drifting sinusoidal gratings presented in the visual field contralateral to the recording site, and SST nRT neurons were activated optogenetically during randomly interleaved trials while recording LFPs and spiking activity in V1 (*Figure 2F–G*, *Figure 2—figure supplement 1*). Visual stimulation alone reduced theta but enhanced beta- and gamma-band LFP power compared with baseline activity (*Figure 2H*). In the presence of visual stimulation, optogenetic activation of SST nRT neurons significantly enhanced theta power while reducing power in other frequency bands (*Figure 2H*, *Table 1*). However, only the reduction of gamma power was consistently observed across all four recorded mice, while other effects were variable across subjects (*Figure 2I*, *Table 1*).

**Table 1.** Results of significance testing across different conditions.

Power amplitudes are in units of 1000 * uV$^2$/Hz (**Figure 2C–I**, **Figure 4B–E**) and firing rates are in Hz (**Figure 2K, L**, **Figure 4F**). BS: broad spiking; NS: narrow spiking.

| | | Still and laser-off | Still and laser-on | Running and laser-off | Running and laser-on | p-Value |
|---|---|---|---|---|---|---|
| **Figure 2C** (n = 111 channels) | θ | 39 ± 1.8 | 27 ± 1.4 | | | 1.2e-9 |
| | β | 64 ± 3.4 | 38 ± 2.3 | | | 4.7e-10 |
| | γ | 131 ± 6.5 | 67 ± 4.4 | | | 2.5e-12 |
| **Figure 2D** (n = 111 channels) | θ | 39 ± 1.8 | | 45 ± 2.8 | | 0.70 |
| | β | 64 ± 3.4 | | 49 ± 2.7 | | 4.5e-5 |
| | γ | 131 ± 6.5 | | 201 ± 9.2 | | 5.2e-12 |
| **Figure 2E** (n = 384 channels, four mice) | θ | 29 ± 7.1 | 1.8e4 ± 5.0 | 33 ± 10.6 | 32 ± 9.5 | 0.12 (still), 0.46 (running) |
| | β | 47 ± 10.0 | 2.7e4 ± 6.3 | 36 ± 6.0 | 32 ± 6.4 | 0.07 (still), 0.69 (running) |
| | γ | 95 ± 18.5 | 5.0e4 ± 10.3 | 137 ± 31.04 | 83 ± 17.7 | 0.009 (still), 0.01 (running) |
| **Figure 2H** (n = 94 channels) | θ | −8.9 ± 0.5 | 16 ± 1.1 | | | 3.8e-17 |
| | β | 18 ± 1.6 | 5.7 ± 1.1 | | | 4.9e-9 |
| | γ | 39 ± 1.4 | −42 ± 1.6 | | | 3.8e-17 |
| **Figure 2I** (n = 322 channels, four mice) | θ | 6.2 ± 8.7 | 1.1e4 ± 7.7 | −0.9 ± 2.8 | 1.9e4 ± 6.4 | 0.71 (still), 0.03 (running) |
| | β | 22 ± 11.9 | 3.3e4 ± 5.9 | 3.7 ± 2.9 | 1.5e4 ± 7.4 | 0.28 (still), 0.64 (running) |
| | γ | 17 ± 7.6 | −2.3e4 ± 7.1 | 19 ± 8.5 | −2.8e4 ± 9.2 | 0.004 (still), 0.005 (running) |
| **Figure 2K** (n = 134 BS, 51 NS, in four mice) | BS | 10.6 ± 1.61 | 6.9 ± 1.00 | 16.9 ± 2.65 | 10.5 ±. 59 | 0.0009 (still), 0.001 (running) |
| | NS | 15.98 ± 2.94 | 9.2 ± 1.33 | 25.9 ± 4.34 | 15.2 ± 2.40 | 0.003 (still), 0.006 (running) |
| **Figure 2L** (n = 100 BS, 32 NS cells, in four mice) | BS | −38.2 ± 0.83 | −23.6 ± 4.72 | −38.2 ± 1.01 | −34.0 ± 4.02 | |
| | NS | −38.5 ± 0.65 | −21.1 ± 8.10 | −38.0 ± 1.17 | −35.56 ± 5.31 | |
| **Figure 4B** (n = 121 channels, in two mice) | θ | 22 ± 1.4 | 22 ± 1.4 | | | 0.78 |
| | β | 42 ± 5.3 | 45 ± 5.5 | | | 0.50 |
| | γ | 77 ± 12.5 | 78 ± 12.4 | | | 0.88 |
| **Figure 4C** (n = 121 channels, in two mice) | θ | 22 ± 1.4 | | 27 ± 2.0 | | 0.04 |
| | β | 42 ± 5.3 | | 42 ± 4.5 | | .49 |
| | γ | 77 ± 12.5 | | 99 ± 12.2 | | 0.004 |
| **Figure 4E** (59 channels, in two mice) | θ | 27 ± 0.3 | 3.3 ± 0.4 | | | 0.45 |
| | β | 4.0 ± 0.8 | 4.7 ± 0.7 | | | 0.15 |
| | γ | 6.6 ± 1.8 | 6.5 ± 1.2 | | | 0.33 |
| **Figure 4F** (n = 31 BS, 43 NS cells, in two mice) | BS | 10.8 ± 1.72 | 11.8 ± 1.81 | 15.8 ± 2.64 | 16.1 ± 2.75 | 0.81 (still), 0.98 (running) |
| | NS | 12.0 ± 2.13 | 12.3 ± 2.15 | 18.9 ± 4.05 | 18.3 ± 3.90 | 0.91 (still), 0.87 (running) |

Given that the reduction in gamma power in response to activation of SST nRT neurons could potentially be caused by reduced spiking of neurons in V1, we next investigated firing rates of isolated neurons in V1. Neurons in V1 were classified as narrow spiking (NS) or broad spiking (BS) using three parameters calculated from average spike waveforms (**Niell and Stryker, 2008**; **Figure 2J**). NS cells consist of fast-spiking interneurons, whereas BS cells are 90% excitatory and 10% inhibitory cells (**Barthó et al., 2004**; **Atencio and Schreiner, 2008**). Optogenetic activation of SST nRT neurons during visual stimulation significantly suppressed activity of both BS and NS cell types in V1 of all four mice during both stationary and locomotion states (**Figure 2K**, **Table 1**).

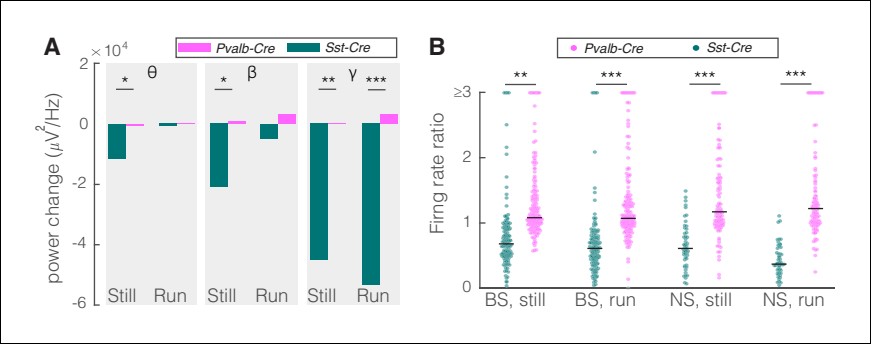

**Figure 3.** Comparison of optogenetic activation of somatostatin (SST) versus parvalbumin (PV) nucleus reticularis thalami (nRT) neurons on V1. (**A**) Changes in the V1 ongoing power in response to optogenetic stimulation of SST (teal) and PV (pink) neurons in nRT (**Table 2**). *p<8e-3, **p<1.6e-3, ***p<1.6e-4. (**B**) Effect of optogenetic activation of SST and PV neurons in nRT on the ratio of laser-on to laser-off firing rates in narrow spiking (NS) and broad spiking (BS) cells in V1 in still and running conditions (SST vs. PV; BS, still: 0.79 vs. 1.27, p=0.0006, run: 0.67 vs. 1.29, p=1e-6, n: 134 vs. 191 cells; NS, still: 0.65 vs. 1.46, p=2e-5, run: 1.46 vs. 1.50, p=8e-7, n: 51 vs. 113; all in four mice). **p<2.5e-3, ***p<2.5e-4.

To test whether different cortical layers are disproportionately modulated by the optogenetic activation of SST nRT neurons, we compared the changes in visual responses at individual recording sites. Visual stimuli evoked strong responses in all layers (**Figure 2—figure supplement 2A, B**), and optogenetic activation of SST nRT neurons reduced the activity evoked by visual stimuli (**Figure 2—figure supplement 2C, D**). Interestingly, the changes in LFP power produced by optogenetic manipulation of the nRT neurons were not solely due to reduction in the firing rate of V1 neurons (**Figure 2L**, **Table 1**).

Given that PV nRT neurons projected less densely to dLGN, we hypothesized that modifying the activity of these neurons would have a smaller, if any, effect on V1. Indeed, consistent with their

**Table 2.** Results of significance testing across different conditions.
Power amplitudes are in units of 1000 * uV²/Hz. PV: parvalbumin; SST: somatostatin.

| | | | SST: laser-on–laser-off | PV: laser-on–laser-off | Mean difference and 95% CI | p-Value |
|---|---|---|---|---|---|---|
| *Figure 3A* (SST: n = 384 channels, four mice; PV: n = 189 channels in four mice) | θ | Still | −11.5 | −1.3 | 10.2, [3.4, 15.2] | 0.005 |
| | | Run | −0.7 | 0.1 | −0.8, [−8.2, 5.1] | 0.14 |
| | β | Still | −20.8 | 4.4 | −25.2, [−29.8, −17.0] | 0.008 |
| | | Run | −4.8 | 2.3 | −7.1, [−10.1, −6.6] | 0.08 |
| | γ | Still | −45.0 | 11.9 | −55.9, [−63.1, −51.4] | 0.001 |
| | | Run | −53.3 | 9.8 | −63.1, [−66.0, −60.2] | 5e-5 |
| *Figure 5A* (SST: n = 313 channels, four mice; PV: n = 121 channels in two mice) | θ | Still | −3.8 | 0.2 | −4.0, [−5.1, 1.3] | 0.72 |
| | | Run | −15.9 | 3.5 | −19.4, [−22.0, −11.1] | 0.06 |
| | β | Still | 6.4 | 4.1 | 2.3, [−0.9, 3.5] | 0.81 |
| | | Run | 3.9 | −0.7 | 4.6, [−2.2, 8.7] | 0.47 |
| | γ | Still | 11.9 | 5.4 | 7.5, [4.1, 10.5] | 0.12 |
| | | Run | 9.8 | 3.8 | 6.0, [3.4, 9.8] | 0.09 |

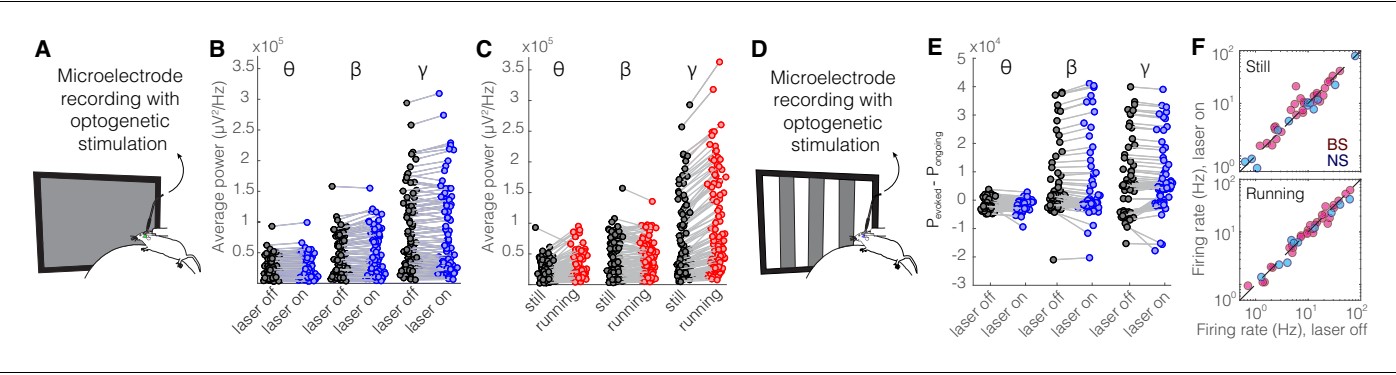

**Figure 4.** Optical stimulation of control enhanced yellow fluorescent protein (eYFP)-expressing somatostatin (SST) nucleus reticularis thalami (nRT) neurons does not change V1 activity. (A) Experimental setup. (B) Across-trial average power of all 121 channels in the absence (black) and presence (blue) of optogenetic activation in one mouse shows no change (*Table 1*). (C) Across-trial average power of all channels in one mouse indicates that locomotion slightly modulates theta and beta powers, while causing a strong enhancement in gamma power (*Table 1*). (D) Visual responses were recorded while mice were presented with moving gratings. (E) Average power across all channels shows that light delivery in eYFP-expressing SST nRT cells does not affect power in V1 (*Table 1*). (F) Firing rate of broad spiking (BS) (magenta) and narrow spiking (NS) (cyan) cells across different conditions is not affected by the light in mice in which nRT does not express the opsin (*Table 1*).

weaker projection to dLGN, activating PV nRT neurons produced only insignificant effects on V1 (*Figure 2—figure supplement 3*, *Figure 2—source data 1*). Surprisingly, the average change in gamma response to PV nRT activation, while not statistically significant, was in the opposite direction than the effect obtained with SST nRT activation (*Figure 3A*, *Table 2*). The difference between effects of activating SST or PV nRT neurons on firing rates was parallel to their effects on gamma (*Figure 3B*).

As a control experiment, we tested for potential non-specific effects of the laser light used for optogenetic activation by performing recordings in the V1 cortex of *Sst-Cre* mice in which an AAV viral construct containing eYFP rather than an opsin was injected into nRT. Visual responses were recorded with and without optogenetic light during interleaved trials (*Figure 4A, D*). As expected, delivering the optogenetic light (blue, 473 nm, ~63 mW/mm$^2$) in nRT had no effect on the power across different frequency bands with (*Figure 4E*, *Table 1*) or without (*Figure 4B*, *Table 1*) visual stimulation; nor did it alter the firing rates of individual neurons (*Figure 4F*, *Table 1*) or the effects of locomotion on LFP power (*Figure 4C*, *Table 1*). These results indicate that the laser light used for optogenetic experiments has no effects in the absence of opsin expression.

## Optogenetic inhibition of SST but not PV nRT neurons enhances single-cell responses in V1

As an alternative approach to determine how nRT gates input to the cortex, we examined the effect of inhibiting either SST or PV nRT neurons by expressing the enhanced *Natronomonas halorhodopsin* (eNpHR) in *Sst-Cre* or *Pvalb-Cre* mice. The outcome of this experiment is not trivial since optogenetic activation and inactivation of neurons do not necessarily produce symmetric effects (*Phillips and Hasenstaub, 2016*; *Moore et al., 2018*). We found that inhibiting eNpHR-expressing SST nRT neurons at baseline increased the firing rates of BS and NS cells in V1 during running but the increase in gamma did not reach significance (*Figure 5*, *Table 2*, *Figure 5—figure supplement 1*, *Figure 2—source data 1*). Inhibiting eNpHR-expressing PV nRT neurons neither significantly changed power in any of the frequency bands nor firing rates during still or running states (*Figure 5*, *Table 2*, *Figure 5—figure supplement 2*, *Figure 2—source data 1*).

## Optogenetic activation of SST nRT but not PV nRT neurons diminishes encoding ability of both BS and NS cells in V1

The action potentials in V1 neurons represent features of the visual world. Information theory provides a means to calculate how action potentials of a neuron inform us about what visual stimulus

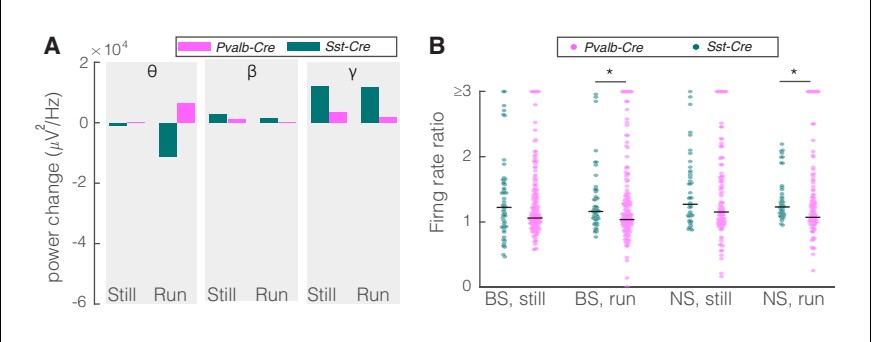

**Figure 5.** Comparison of optogenetic inhibition of somatostatin (SST) versus parvalbumin (PV) nucleus reticularis thalami (nRT) neurons on V1. (**A**) Changes in the V1 ongoing power due in response to optogenetic inhibition of SST (teal) and PV (pink) neurons in nRT across the three frequency bands (**Table 2**). (**B**) Effect of the optogenetic inhibition of SST and PV neurons in nRT on the ratio of laser-on to laser-off firing rates in narrow spiking (NS) and broad spiking (BS) cells in V1, in still and running conditions during visual stimulation (SST vs. PV; BS, still: 1.33 vs. 1.27, p=0.05, run: 1.29 vs. 1.10, p=0.004, n: 58 [in four mice] vs. 127 [in two mice] cells; NS, still: 1.47 vs. 1.26, p=0.13, run: 1.35 vs. 1.17, p=0.01, n: 44 vs. 75). *p<0.0125, **p<2.5e-3.

The online version of this article includes the following figure supplement(s) for figure 5:

**Figure supplement 1.** Optogenetic inhibition of somatostatin (SST) nucleus reticularis thalami (nRT) neurons increases gamma activity in V1 during visual stimulation.

**Figure supplement 2.** Optogenetic inhibition of parvalbumin (PV) nucleus reticularis thalami (nRT) neurons does not significantly change gamma activity in V1.

the animal was viewing. Given that SST nRT neurons are well positioned to modulate the information transmitted through the dLGN to V1, we asked, quantitatively, to what extent the activity of the SST nRT neurons modifies the encoding of visual information in V1. We computed the mutual information (I(R, S), see Materials and methods) conveyed by the spikes of single neurons about the visual stimuli that gave rise to those responses. We did this with and without optogenetic activation of SST nRT neurons, and separately for the running and stationary behavioral states, because locomotion alters both strength and information content of visual responses (*Niell and Stryker, 2008*; *Dadarlat and Stryker, 2017*). Optogenetic activation of SST nRT neurons markedly reduced mutual information between the neuronal responses and our set of visual stimuli for both BS and NS cells in V1 (*Figure 6A, D*, *Figure 6—figure supplement 1A, B*). The reduction of mutual information in individual V1 neurons during optogenetic activation of SST nRT neurons suggests that activity of the V1 population as a whole would encode less information about visual stimuli. We estimated the representation of visual information in the response of the V1 population by training a linear decoder (LDA) to identify the visual stimulus that the animal was viewing in single stimulus trials on the basis of the spike responses in the entire population of recorded neurons. LDA incorporates the following three assumptions: that different visual stimuli evoke linearly separable responses, that evoked responses are independent across neurons, and that the responses have a Gaussian distribution. The decoder is trained on all data except a single trial, which is left out for testing purposes (leave-one-out cross validation [LDA-LOOXV]). This approach allows us to quantify how well orientation of the visual stimulus can be predicted for the single trials excluded from the training set.

Single-trial neuronal responses during locomotion were classified less accurately during optogenetic activation of SST nRT neurons for grating orientation in both BS (98% laser-off vs. 81% laser-on, p=2e-11, Wilcoxon rank-sum test) and NS (86% vs. 58%, p=1e-13) cells (*Figure 6B, E*). This finding indicates that the cortical representation of information about the visual world is reduced when SST nRT neurons are active. Importantly, this finding did not depend on the behavioral state of the animal (*Figure 6—figure supplement 1C*). Moreover, repeating the decoding analysis separately including only cells that are in the same range of cortical depth yielded similar accuracy (data not shown), indicating no particular laminar distinction in stimulus encoding.

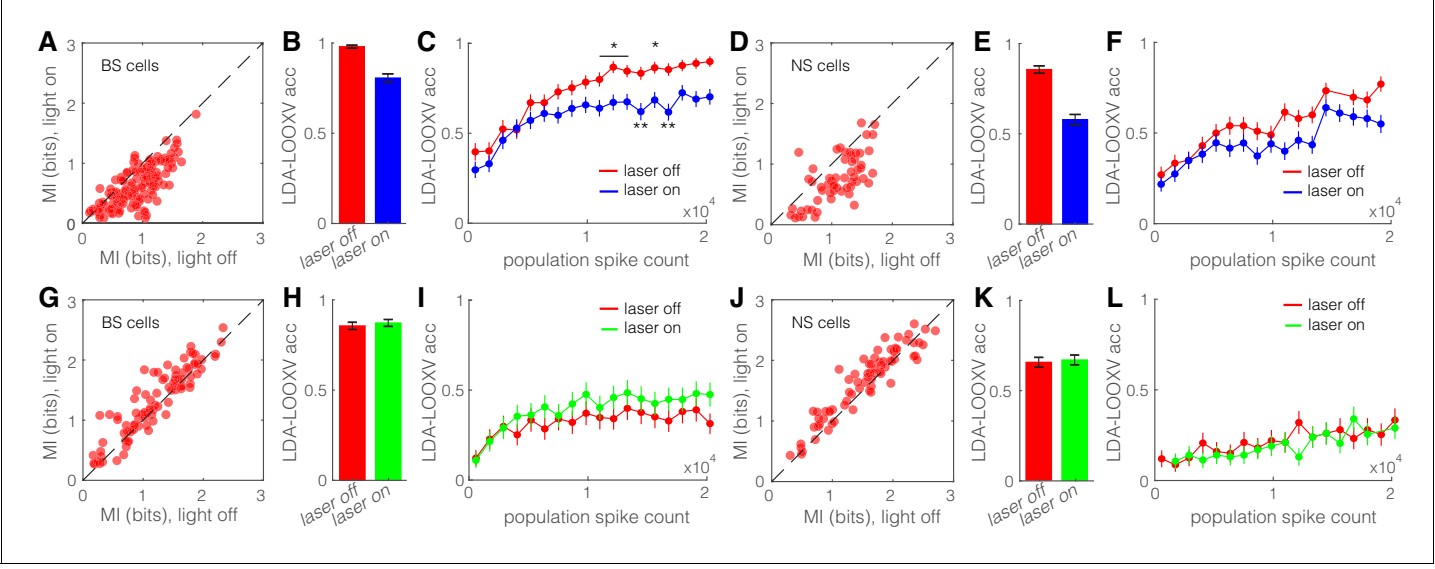

**Figure 6.** Optogenetic activation of somatostatin (SST) nucleus reticularis thalami (nRT) neurons diminishes encoding ability in both broad spiking (BS) and narrow spiking (NS) cells in V1. (**A**) Effects of optogenetic activation of SST nRT cells on BS single-cell mutual information (MI) during locomotion: single-cell MI of BS neurons during locomotion demonstrates a significant reduction with optogenetic activation of SST nRT neurons (MI: light-off 0.91 ± 0.03 to light-on 0.64 ± 0.03, p=6e-5, n = 141 cells, four mice). Dashed line indicates unity. (**B**) Accuracy in leave-one-out cross validation (LDA-LOOXV) classification of visual stimulus movement orientation using BS responses during locomotion and optogenetic activation of SST nRT cells: (light-off 0.98 to light-on 0.81, p=2e-11, four mice, Wilcoxon rank-sum test). Error bars indicate bootstrapped estimates of SE. (**C**) Classification accuracy for grating movement orientation as a function of BS population spike count during locomotion and optogenetic activation of SST nRT neurons. Error bars indicate bootstrapped estimates of SE. Chance level would be at 0.16. (**D**) Effects of optogenetic activation of SST nRT cells on NS single-cell MI during locomotion: (MI: light-off 1.09 ± 0.05 to light-on 0.76 ± 0.05, p=7e-4, n = 58 cells, four mice). (**E**) Accuracy in LDA-LOOXV classification of visual stimulus movement orientation using NS responses during locomotion and optogenetic activation of SST nRT cells: (light-off 0.86 to 0.58, p=1e-13, four mice). (**F**) Classification accuracy for grating movement orientation as a function of NS population spike count during locomotion and optogenetic activation of SST nRT neurons. (**G**) Effects of optogenetic inhibition of SST nRT cells on BS single-cell MI during locomotion: (1.19 ± 0.06 to 1.31 ± 0.06, p=2e-4, n = 90 cells, four mice). (**H**) Accuracy in LDA-LOOXV classification of visual stimulus movement orientation using BS responses during locomotion and optogenetic inhibition of SST nRT cells: (light-off 0.86–0.87, p=0.53, four mice). (**I**) Classification accuracy for grating movement orientation as a function of BS population spike count during locomotion and optogenetic inhibition of SST nRT neurons. (**J**) Effects of optogenetic inhibition of SST nRT cells on NS single-cell MI during locomotion: (MI: 1.54 ± 0.07 to 1.65 ± 0.06, p=7e-4, n = 74 cells, four mice). (**K**) Accuracy in LDA-LOOXV classification of visual stimulus movement orientation using NS responses during locomotion and optogenetic inhibition of SST nRT cells: (light-off 0.65–0.67, p=0.72, four mice). (**L**) Classification accuracy for grating movement orientation as a function of NS population spike count during locomotion and optogenetic inhibition of SST nRT neurons. *p<0.05, **p<0.01.

The online version of this article includes the following figure supplement(s) for figure 6:

**Figure supplement 1.** Optogenetic manipulation of somatostatin (SST) nucleus reticularis thalami (nRT) neurons alters encoding accuracy in both broad spiking (BS) and narrow spiking (NS) cells.

**Figure supplement 2.** Optogenetic perturbation of parvalbumin (PV) nucleus reticularis thalami (nRT) neurons does not alter encoding accuracy.

## Reduction in V1 information produced by nRT activation is not solely due to the reduction in firing rate

Optogenetic activation of SST nRT neurons led to lower population spike counts on average, which in turn led to reduced visual information (*Figure 6—figure supplement 1A, B*). The observed reduction in visual information could therefore be due either to the reduction of neuronal firing rates or to changes in the pattern of neural responses. To distinguish between these two possibilities, we quantified decoding accuracy for trials with equal population spike counts (see Materials and methods). We sampled (without replacement) anywhere between one and maximum number of neurons to get a very low or high number of spikes, respectively. This process was repeated 70 times, and then decoding accuracy was compared for equal population spike counts during laser-off and laser-on states by including more cells in the laser-on than the laser-off classifier. LDA-LOOXV was performed separately for the data collected from each mouse, and results from all four mice were pooled to

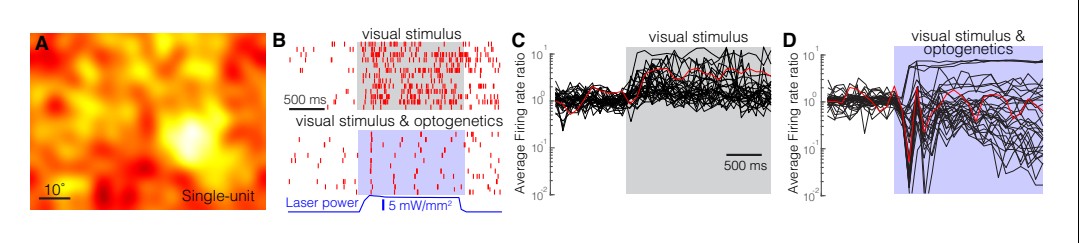

**Figure 7.** Optogenetic activation of somatostatin (SST) nucleus reticularis thalami (nRT) neurons reduces firing in dorsal lateral geniculate nucleus (dLGN) neurons. Spike-triggered averages of a single unit (A) in dLGN were calculated using responses to low-frequency filtered noise stimulus. (B) Spike raster for a representative single unit in dLGN across several trials (rows) with visual stimulus only (top) or visual stimulus coupled with optogenetic activation of SST nRT neurons (bottom). Gray shaded area shows the duration of the visual stimulus. Blue shading shows the duration of optogenetic activation. Average firing rate ratios (computed as the visually evoked firing rates for the optimal grating divided by the firing rates prior to the onset of visual stimulation) of dLGN single units in response to visual stimulation only (C) and in response to visual stimulation coupled with optogenetic activation of SST nRT neurons (D). Red trace in (C) and (D) is the example cell shown in B.

The online version of this article includes the following video, source data, and figure supplement(s) for figure 7:

**Source data 1.** Results of significance testing across different conditions.

**Figure supplement 1.** Optogenetic activation of somatostatin (SST) nucleus reticularis thalami (nRT) neurons significantly reduces gamma activity in dorsal lateral geniculate nucleus (dLGN) both with and without visual stimulation.

**Figure supplement 2.** Optogenetic inhibition of somatostatin (SST) nucleus reticularis thalami (nRT) neurons significantly enhances gamma activity in dorsal lateral geniculate nucleus (dLGN) both with and without visual stimulation.

**Figure supplement 3.** Optogenetic perturbation of somatostatin (SST) nucleus reticularis thalami (nRT) neurons diminishes spiking reliability.

**Figure 7—video 1.** Optogenetic activation of somatostatin (SST) nucleus reticularis thalami (nRT) neurons reduces the firing in dorsal lateral geniculate nucleus (dLGN) neurons.

https://elifesciences.org/articles/61437#fig7video1

generate average decoding accuracy as a function of population spike count for each cell type, behavioral state, and optogenetic state. Not surprisingly, classification accuracy increased with increasing spike count across all conditions. However, accuracy was always lower during optogenetic activation of SST nRT neurons for equal population spike counts (*Figure 6C, F*, *Figure 6—figure supplement 1D*), and particularly so at high population spike counts. The difference was less prominent with optogenetic inhibition of SST nRT neurons (*Figure 6G–L*, *Figure 6—figure supplement 1E–H*) and optogenetic activation or inhibition of PV nRT neurons (*Figure 6—figure supplement 2*). These findings indicate that optogenetic activation of SST nRT neurons not only reduces spiking activity in V1 cells but also alters the spiking pattern, leading to a less accurate encoding of visual stimuli.

## Optogenetic activation of SST nRT neurons reduces single-cell responses and gamma power in dLGN

Optogenetic activation of the inhibitory neurons in the nRT, including all of the subtypes, transiently reduces activity of thalamocortical neurons in dLGN (*Olsen et al., 2012*; *Reinhold et al., 2015*). Similar results were obtained when optogenetically activating SST nRT neurons in transgenic mice (*Campbell et al., 2020*). To determine how effective activation of only the SST nRT subpopulation is in reducing activity in dLGN, we expressed ChR2 in SST nRT neurons by intra-nRT viral injections in *Sst-Cre* mice and performed single-unit (SU) and multiunit (MU) and field potential recordings from dLGN. Responses to low-frequency filtered noise stimulus were used to calculate spike-triggered averages (STAs, *Figure 7A*). STAs at 18% (35/193) of the recording sites displayed a classical center-surround structure like that shown for a SU (*Figure 7A*). Nearly all of the SU and MU recordings were effectively inhibited during the 4 s optogenetic activation of SST nRT neurons (*Figure 7—video 1*), and field potentials recorded in dLGN showed a dramatic reduction across all frequency bands (*Figure 7—figure supplement 1*, *Figure 7—source data 1*), in agreement with a strong projection of SST nRT neurons to dLGN (see *Figure 1*; *Campbell et al., 2020*). While visual stimuli evoked strong responses in SUs (*Figure 7B*, top), optogenetic activation markedly reduced those responses,

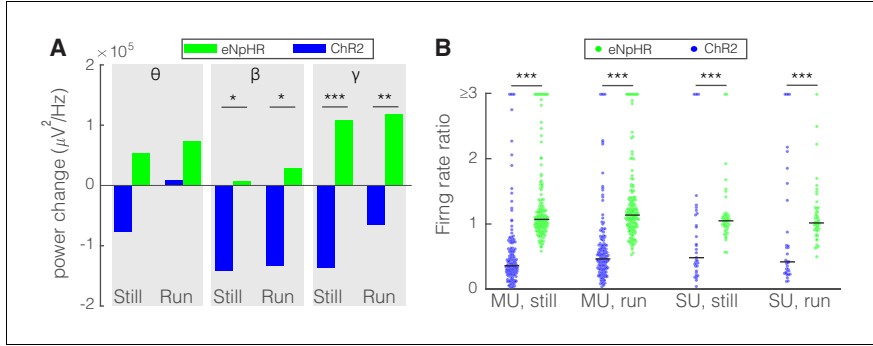

**Figure 8.** Optogenetic activation and inhibition of somatostatin (SST) nucleus reticularis thalami (nRT) neurons bidirectionally control the activity of dorsal lateral geniculate nucleus (dLGN) neurons. (**A**) Changes in the dLGN ongoing power due to optogenetic activation (blue) or inhibition (green) of SST nRT neurons across the three frequency bands (Channelrhodopsin-2 [ChR2] vs. enhanced *Natronomonas halorhodopsin* [eNpHR]; theta, still: −77.6e3 vs. 53.4e3, p=0.06, nested t-test, mean difference −132.0e3 with a 95% CI of [−105.1e3, 163.0e3], run: 8.2e3 vs. 74.1e3, p=0.09, nested t-test, mean difference −65.9e3 with a 95% CI of [−75.1e3, −51.8e3]; beta, still: −141.3e3 vs. 6.7e3, p=0.012, mean difference −134.6e3 with a 95% CI of [−178.0e3, −87.2e3], run: −134.0e3 vs. 28.5e3, p=0.009, mean difference −162.5e3 with a 95% CI of [−184.0e3, −123.6e3]; gamma, still: −137.0e3 vs. 109.3e3, p=3e-5, mean difference −246.3e3 with a 95% CI of [−289.5e3, −191.7e3], run: −65.0e3 vs. 118.9e3, p=6e-3, mean difference −183.9e3 with a 95% CI of [−202.0e3, −58.8e3]; all in three mice, all powers in uV2/Hz). *p<0.016, **p<0.0016, ***p<0.00016. (**B**) Effects of optogenetic activation or inhibition of SST neurons on the ratio of laser-on to laser-off firing rates of dLGN cells across locomotion conditions (ChR2 vs. eNpHR; multi-unit [MU], still: 0.65 vs. 1.28, p=5e-22, run: 0.66 vs. 1.65, p=5e-11, n: 158 vs. 185 cells; single-unit [SU], still: 1.13 vs. 1.30, p=7e-6, run: 1.34 vs. 1.23, p=6e-8, n: 35 vs. 49; all in three mice). ***p<2.5e-4.

---

leaving only a few spikes in response to each cycle of the grating stimulus (sinusoidal drifting gratings, 2 Hz temporal frequency, 0.04 Hz spatial frequency) (*Figure 7B–D*). MU recording sites displayed a similar degree reduction (*Figure 7—figure supplement 1*).

Optogenetic inhibition of SST nRT neurons, in line with our V1 recordings, caused only a slight, and not significant, increase in gamma activity and spiking in dLGN compared to baseline (*Figure 7—figure supplement 2*, *Figure 7—source data 1*).

To test how nRT output alters the reliability of dLGN responses to visual stimulation, we calculated the coefficient of variation (CV) by dividing the standard deviation of firing rates in the presence or absence of optogenetic manipulation by their respective means. Our data reveal that optogenetic activation of SST nRT neurons increased the CV of dLGN responses (*Figure 7—figure supplement 3A*). These findings reveal that the firing of SST nRT neurons reduces the ability of dLGN neurons to reliably report visual information. In contrast, optogenetic inhibition of SST nRT neurons did not significantly reduce CV of dLGN responses (*Figure 7—figure supplement 3B*).

The effects of perturbing SST nRT activity by optogenetics on field potentials recorded in the dLGN were measured in the absence of visual stimulation. As expected, optogenetic excitation generally reduced their amplitude and optogenetic inhibition increased them or left them unchanged. These opposite-directed changes were significant in the beta and gamma bands both during locomotion and at rest (*Figure 8A*). During visual stimulation, which increases the firing of dLGN neurons, activation of SST neurons in nRT produced much stronger effects on dLGN firing of both SU and MU sites than did inhibiting them (*Figure 8B*). These findings from optogenetic perturbation of SST nRT neurons on dLGN activity reveal that activating nRT neurons powerfully reduces the visual input to V1.

## Discussion

Here, we have identified new specificity in a subcortical circuit that modulates both gamma power and the representation of information in the discharge of neurons in the primary visual cortex. We found that the input from nRT to the dLGN in the mouse is predominantly from the class of

inhibitory neurons that express SST, with negligible anatomical connections from the more numerous PV-positive neurons (*Figure 1*, *Figure 1—figure supplement 1*). Optogenetic activation of SST nRT neurons in alert, head-fixed mice suppressed the SU spiking and field potential responses to visual stimulation in both dLGN and V1, and caused suppression of gamma, whether the mice were stationary or running on a polystyrene ball floating on air. Inhibition of SST nRT neurons produced mostly opposite effects (*Figures 3* and *5*). Perturbation of PV neurons in nRT had much smaller effects compared to SST nRT neurons, consistent with fewer projections to dLGN (*Figures 3* and *5*). These findings provide evidence for a specific neural circuit that regulates gamma power, and which is associated with visual attention, and the encoding of visual information in V1.

## Cellular heterogeneity in the nRT

SST and PV neurons have been defined by antibody staining of the protein markers, by genetic labeling throughout development, and by their recombinase activity in adulthood. These three approaches to distinguishing SST from PV cell types may differ in the extent of overlap that is interpreted as coexpression. The findings of this paper are based on the third approach and do not bear on the issue of coexpression assessed from the other approaches. The present study reveals the distinct functional effects of *Sst-Cre* and *Pvalb-Cre* expressing neurons.

The existence of functionally distinct neuron types within the nRT has been shown (*Lee et al., 2007*; *Halassa et al., 2014*) and has recently been associated with PV and SST markers (*Clemente-Perez et al., 2017*) and further extended to additional molecular markers (*Li et al., 2020*; *Martinez-Garcia et al., 2020*). Finding molecular markers of functionally distinct nRT cells may be significant because it could allow us to selectively perturb distinct functions of the nRT.

Our studies used virally injected constructs in mature *Pvalb-Cre* and *Sst-Cre* mice to distinguish PV and SST neurons in the various sectors of the nRT (*Clemente-Perez et al., 2017*), including in the visual sector (*Figure 1—figure supplement 1*; *Clemente-Perez et al., 2017*). Using immunohistochemistry and confocal imaging, we confirmed that putative synaptic boutons from PV and SST nRT neurons are largely distinct (*Figure 1—figure supplement 2*), although certain neurons do express both markers (*Figure 1—figure supplement 2*) consistent with previous work (*Clemente-Perez et al., 2017*). Calbindin has been suggested as another marker of a subpopulation of nRT neurons (*Martinez-Garcia et al., 2020*); however, this has not been confirmed by single-nuclear RNA sequencing studies, which found no absolutely exclusive genetic markers between nRT cell types, although gradients in *Spp1* and *Ecl1* gene expression were noted between core and shell parts of the nRT (*Li et al., 2020*). Future studies at multiple levels of analysis (protein, genetic) will determine whether the differences between results of the studies cited above are due to use of different transgenic mouse models. In the present study, using viral expression of fluorescent reporters of recombination in adult mice, we find that both PV and SST neurons are present in the visual sector of the nRT (*Figure 1—figure supplements 1* and *2*) and that both types project to visual thalamic nuclei, but the SST neurons predominate in the projection to dLGN, whereas PV neurons project more strongly to higher-order visual nuclei (LP) (*Figure 1*). The role of neurons that express both markers (PV and SST) and their specific contribution remains to be determined; however, our results demonstrate that the functional effects of *Sst-Cre* and *Pvalb-Cre* neurons in nRT are distinct.

Notably, the studies cited above were all done in mice, which are more readily amenable to molecular studies than other species. Such markers will need to be validated in other mammals, and non-human primates and humans in future studies. However, the existence of mutually exclusive PV and SST neurons in the adult human nRT (*Clemente-Perez et al., 2017*) suggests that our findings in mice may be relevant to the human brain.

## Comparison with carnivore and primate

The perigeniculate nucleus in the carnivore and primate is thought to be the portion of the nRT related to the thalamocortical visual system, although this is contested (*Ahlsén et al., 1982*). The carnivore perigeniculate consists of inhibitory neurons that receive excitatory input from ascending thalamocortical axons of the dLGN as well as from layer 6 cells of V1, and they project back to inhibit the principal cells of the dLGN in a highly focused topographic fashion (*Lam and Sherman, 2011*; *Soto-Sánchez et al., 2017*; *Crabtree, 2018*). This focal projection of the nRT is consistent with the 'Searchlight' hypothesis for nRT function (*Crick, 1984*). It is not known whether the nRT projection

to the mouse dLGN has sufficiently precise topography to play such a focal role in directing attention to particular areas of the field. It is possible that the portion of the carnivore nRT referred to by *Ahlsén et al., 1982* as 'reticular neurons' may be analogous or even homologous to the nRT of the mouse. Such an arrangement could be consistent with a role for the mouse nRT in switching attention between modalities (*Wimmer et al., 2015*) rather than among different loci in the visual field. However, recent findings indicate a more focal role for the mouse nRT in corticothalamic feedback that modifies the properties of receptive fields in dLGN (*Born et al., 2020*), consistent with the focal retrograde labeling we observed in *Figure 1—figure supplement 2*.

Earlier studies have found that, as in the mouse, the nRT of cats (*Oertel et al., 1983*; *Clemence and Mitrofanis, 1992*), ferrets (*Clemence and Mitrofanis, 1992*), and monkeys (*Graybiel and Elde, 1983*; *Jones and Hendry, 1989*) contains both PV and SST cells. In those studies, the projections of the specific cell types to visual thalamic nuclei and their roles in the transmission of visual information were not determined. Interestingly, in one primate species, the *Galago senegalensis*, different laminae of the visual portion of the nRT project to the dLGN and the pulvinar, which corresponds loosely to the mouse LP (*Conley and Diamond, 1990*). Cells in the more medial laminae, which is rich in SST cells, project to the pulvinar, while cells in the more lateral laminae project to the dLGN (*Conley et al., 1991*). These findings suggest a functional segregation of the roles of SST and PV cells in this species, although in the opposite direction to that found in mice, where the dLGN receives a more substantial projection from SST cells and LP a denser projection from PV cells.

## Does nRT activity merely turn off the input to V1?

Activation of SST nRT neurons caused a robust reduction in the firing of both the thalamocortical neurons in dLGN and the excitatory and putative fast spiking inhibitory interneurons in V1 (*Figures 2*, *7,* and *8*). Consistent with this effect, electron microscopy reveals geniculate terminations of SST nRT neurons only on dLGN excitatory neurons (*Campbell et al., 2020*). Interestingly, activating SST nRT neurons alters not just the amount but also the pattern of activity across the population in V1. In contrast, despite the fact that PV neurons are present in the visual portion of the nRT (see *Figure 1— figure supplement 1*), perturbation of their firing rate did not affect the visual information encoding in V1. Optogenetic perturbation of PV nRT neurons had a smaller effect on V1 compared with SST nRT neurons, and some of these effects were in the opposite direction. For instance, activation of PV nRT neurons increased gamma activity and firing rates in V1 cells while SST nRT neurons did the opposite (*Figure 3*). These findings are consistent with the possibility that the PV nRT neurons inhibit dLGN interneurons and/or SST nRT neurons, although such connections are yet to be established. We propose that the different outputs of SST and PV nRT neurons to primary and high-order visual thalamic nuclei, and the possible intra-nRT connections, may support the switching of attention as suggested for the nRT (*Wimmer et al., 2015*).

We speculate that the more potent effects of transiently silencing V1 by the activation of all nRT neurons in experiments using *Gad2-Cre* mice are due to the simultaneous suppression of both dLGN and LP thalamocortical neurons (*Reinhold et al., 2015*). Interestingly, excitation of SST nRT neurons does not silence activity in V1 but only reduces the information about the visual world that V1 carries.

SST and PV neurons in the somatosensory sector of the nRT have distinct burst firing properties *Clemente-Perez et al., 2017*; whether this is also true in the visual sector remains to be determined. Distinct bursting properties have been demonstrated in the rat visual nRT (*Kimura et al., 2012*), but it remains unknown whether these are associated with distinct molecular markers.

## Cortical VIP neurons also modulate V1 activity

Locomotion increases both gamma power and visual responses in mouse V1, and gates a form of adult plasticity (*Niell and Stryker, 2010*; *Kaneko and Stryker, 2014*; *Kaneko et al., 2017*; *Hoseini et al., 2019*). The effects of locomotion are produced by a circuit operating through vasoactive intestinal peptide (VIP) interneurons in V1 (*Fu et al., 2014*; *Fu et al., 2015*). During locomotion, when gamma power is strong in dLGN and V1, activation of SST nRT neurons reduces it (*Figures 2*, *3,* and *8*). Visual responses of both excitatory and inhibitory cortical neurons were reduced to a similar extent by activation of SST nRT cells (*Figure 2*). These findings indicate that these two

modulatory systems—locomotion via cortical VIP cells (*Niell and Stryker, 2010*; *Fu et al., 2014*) and SST nRT activation—contribute independently to activity in V1. In the somatosensory representation, SST nRT neurons receive inputs from mainly subcortical structures (central amygdala, anterior thalamus, external segment of globus pallidus) in contrast with PV nRT neurons that mainly receive sensory cortical inputs (*Clemente-Perez et al., 2017*). We speculate that the SST nRT neurons are well positioned to exert a bottom-up regulation of visual attention. In contrast, the effects of locomotion on V1 activity are regulated by cortical VIP interneurons that receive top-down inputs from higher cortical areas (*Zhang et al., 2014*).

### Implications for disease

Sensory stimulation in the gamma range has been shown to enhance cognition in a mouse model of Alzheimer's disease (*Adaikkan et al., 2019*). Given that nRT is involved in sensory processing and attention, and that its dysfunction has been implicated in attention disorders (*Zikopoulos and Barbas, 2012*; *Ahrens et al., 2015*; *Wells et al., 2016*), and given that gamma power has been associated with attention (*Kim et al., 2016*), we propose that SST nRT neurons could be a target for modulating visual attention.

# Materials and methods

**Key resources table**

| Reagent type (species) or resource | Designation | Source or reference | Identifiers | Additional information |
|---|---|---|---|---|
| Strain, strain background (B6.129P2 Pvalbtm1(cre)Arbr/J) *Mus musculus* Male and female | *Pvalb-Cre* | The Jackson Laboratory | Stock No: 017320 RRID:IMSR_JAX:017320 | |
| Strain, strain background (B6N.Cg-Ssttm2.1(cre)Zjh/J) *Mus musculus* Male and female | *Sst-Cre* | The Jackson Laboratory | Stock No: 018973 RRID:IMSR_JAX:018973 | |
| Strain, strain background (C57BL/6J) *Mus musculus* Male and female | C57BL/6J | The Jackson Laboratory | Stock No: 000664 RRID:IMSR_JAX:000664 | |
| Sequence-based reagent | Pvalb-1 WT/Pvalb-5 KO | Transnetyx | Genotyping probes | |
| Sequence-based reagent | Sst-1 WT/Sst-5 Tg | Transnetyx | Genotyping probes | |
| Viral prep (rAAV5/EF1a-EYFP) | eYFP | Addgene | Addgene plasmid # 27056 SCR_002448 RRID:Addgene_27056 | |
| Viral prep (rAAV5/EF1a-ChR2-EYFP) | ChR2 | Addgene | Addgene plasmid # 20298 RRID:Addgene_20298 | |
| Viral prep (rAAV5/EF1a-eNpHR3.0-EYFP) | eNpHR | Addgene | Addgene plasmid # 26966 RRID:Addgene_26966 | |
| Immunohistochemistry reagent | Normal donkey serum | Jackson Immunoresearch | Cat #: 017-000-121 RRID:AB_2337258 | |
| Immunohistochemistry reagent | Antifade medium (Vectashield) | Vector Laboratories | Cat #: H-1000 RRID:AB_2336789 | |
| Immunohistochemistry reagent | Cholera Toxin Subunit B (Recombinant), Alexa Fluor 488 Conjugate | Invitrogen | Cat #: C34775 | |
| Antibody | Rabbit anti-parvalbumin antibody | Swant | Cat #: PV27 RRID:AB_2631173 | 1:500 dilution |

*Continued on next page*

*Continued*

| Reagent type (species) or resource | Designation | Source or reference | Identifiers | Additional information |
|---|---|---|---|---|
| Antibody | Donkey anti-rabbit Alexa Fluor 594 | Abcam | Cat #: ab150076 RRID:AB_2782993 | 1:500 dilution |
| Antibody | Rabbit anti-somatostatin antibody, unconjugated | Peninsula Laboratories International | Cat #: T-4103 RRID:AB_518614 | 1:1000 dilution |
| Antibody | Monoclonal anti parvalbumin antibody | Swant | Cat #: 235 RRID:AB_10000343 | 1:1000 dilution |
| Antibody | Donkey anti-Mouse IgG (H+L) highly cross-adsorbed secondary antibody, Alexa Fluor 555 | Thermo Fisher Scientific | Cat# A-31570 RRID:AB_2536180 | |
| Surgery item | Metabond | PATTERSON DENTAL / Parkell Co. | Three items: Cat #: 5533559 Cat #: 5533500 Cat #: 5533492 | |
| Equipment | Fiber optic | ThorLabs | Cat #: CFML12U-20 | |
| Equipment | Green laser for eNpHR | OPTO ENGINE LLC | Cat #: MGL-III-532 | |
| Equipment | Blue laser for ChR2 | OPTO ENGINE LLC | Cat #: MDL-III-450 | |
| Software | MATLAB | MathWorks | SCR_001622 | |
| Software | GraphPad Prism 8 | GraphPad | SCR_002798 | |
| Software | Origin 9.0 | OriginLab | SCR_002815 | |
| Software | R | R-project | SCR_001905 | |
| Software | SigmaPlot | SigmaPlot | SCR_003210 | |

## Animals

We performed all experiments in compliance with protocols approved by the Institutional Animal Care and Use Committees at the University of California, San Francisco, and Gladstone Institutes (protocol numbers AN180588-02C and AN174396-03E). Precautions were taken to minimize stress and the number of animals used in all experiments. We followed the NIH guidelines for rigor and reproducibility of the research. Adult (P60–P180) male and female mice of the following genotypes were used: *Sst-Cre* mice, *Pvalb-Cre* mice, and C57BL/6J mice.

## Viral delivery in nRT for optogenetic experiments

We performed stereotaxic injections of viruses into the nRT as described (*Paz et al., 2011*; *Paz et al., 2013*; *Clemente-Perez et al., 2017*; *Ritter-Makinson et al., 2019*). We targeted the nRT with the following stereotaxic coordinates: 1.3 mm posterior to bregma and 2.0–2.1 mm lateral to the midline at two different injection depths (200 nl at 2.65 and 200 nl at 3.0 mm) ventral to the cortical surface. We previously validated that this protocol results in specific expression of the viral construct in the nRT neurons and not in the surrounding brain areas (*Clemente-Perez et al., 2017*) (see also *Figure 1—figure supplement 1*). For eYFP tracing studies (*Figure 1*), a total of 400 nl of concentrated virus ($2 \times 10^{12}$ genome copies per milliliter) carrying genes for eYFP alone (rAAV5/EF1a-EYFP) were injected unilaterally in nRT of *Pvalb-Cre* and *Sst-Cre* mice. For optogenetic experiments, a total of 400 nl of concentrated virus ($2 \times 10^{12}$ genome copies per milliliter) carrying genes for ChR2 or eNpHR were injected unilaterally in nRT of *Pvalb-Cre* and *Sst-Cre* mice, as described (*Clemente-Perez et al., 2017*). The Allen Brain and the Paxinos mouse brain atlases were used to validate the location of the viral expression in nRT (*Paxinos and Franklin, 2001*; *Lein et al., 2007*).

## Headplate surgery and implanting fiber optic

Three to six weeks after viral injections in the nRT, we performed a second surgery to implant a fiber optic (core diameter, 200 μm) in nRT at 1.3 mm posterior to bregma, 1.9–2.0 mm lateral to the midline, and 2.3–2.5 mm ventral to the cortical surface; and a titanium headplate—circular center with a 5 mm central opening—above the V1 cortex (−2.9 mm posterior to bregma, 2.5 mm lateral to the midline) or dLGN (−2.0 mm posterior to bregma, 2.0 mm lateral to the midline). The base of the

fiber optic and the entire skull, except for the region above V1 or dLGN, was covered with Metabond. One week after the recovery from this surgery, the animal was allowed to habituate to the recording setup by spending 15–30 min on the floating ball over the course of 1–3 days, during which time the animal was allowed to run freely. About 2 weeks following this surgery (i.e., ~4–6 weeks after viral injection in nRT), the animal's head was fixed to a rigid crossbar above a floating ball. The polystyrene ball was constructed using two hollow 200-mm-diameter halves (Graham Sweet Studios) placed on a shallow polystyrene bowl (250 mm in diameter, 25 mm thick) with a single air inlet at the bottom. Two optical USB mice, placed 1 mm away from the edge of the ball, were used to sense rotation of the floating ball and transmit signals to our data analysis system using custom driver software. These measurements are used to divide data into still and running trials and analyze them separately.

## Microelectrode recordings in alert mice

To control for circadian rhythms, we housed our animals using a fixed 12 hr reversed light/dark cycle and performed recordings between roughly 11:00 AM and 6:00 PM. All the recordings were made during wakefulness in awake, head-fixed mice that were free to run on the floating ball (*Figure 2A*; *Hoseini et al., 2019*). On the day of recording, the animal was anesthetized with isoflurane (3% induction, 1.5% maintenance) and a craniotomy of about 1 mm in diameter was made above V1 or dLGN. After animals recovered from anesthesia for at least 1 hr, a 1.1-mm-long double-shank 128-channel electrode (*Du et al., 2011*), fabricated by the Masmanidis laboratory (University of California, Los Angeles) and assembled by the Litke laboratory (University of California, Santa Cruz), was slowly inserted through the cranial window. To record from V1, the electrode was placed at an angle of 20–40° to the normal of the cortical surface and inserted to a depth of ~1000 µm. To record from dLGN, the electrode was placed at a normal angle to the cortical surface and inserted to a depth of 2.5–3.0 mm (*Piscopo et al., 2013*). An optical fiber (200 µm diameter) coupled to a light source (green laser for eNpHR, peak intensity ~104 mW/mm$^2$ at 532 nm; blue laser for ChR2, peak intensity ~63 mW/mm$^2$ at 473 nm) was connected to the implanted fiber optic in order to deliver light into nRT. Laser power (3–20 mW) was measured at the end of the optical fiber before connecting to the animals. Recordings were started an hour after electrode insertion.

## Visual stimuli

Stimuli were displayed on an LCD monitor (Dell, 30 × 40 cm, 60 Hz refresh rate, 32 cd/m$^2$ mean luminance) placed 25 cm from the mouse and encompassing azimuths from −10° to 70° in the contralateral visual field and elevations from −20° to +40°. In the first set of recordings, no stimulus was presented (uniform 50% gray) while nRT was exposed to the optogenetic light for 4 s every 20 s. For the second set of recordings, drifting sinusoidal gratings at eight evenly spaced directions (20 repetitions, 2 s duration, 0.04 cycles per degree, and 1 Hz temporal frequency) were generated and presented in random sequence using the MATLAB Psychophysics Toolbox (*Brainard, 1997*; *Kleiner et al., 2007*), followed by 2 s blank period of uniform 50% gray. This stimulus set was randomly interleaved with a similar set in the presence of optogenetic light. Optogenetic stimulation was delivered for 2 s periods beginning simultaneously with the onset of the visual stimulus, overlapping the entire stimulus period and turned off by the end of the stimulus.

## Data acquisition and analysis

Movement signals from the optical mice were acquired in an event-driven mode at up to 300 Hz and integrated at 100-ms-long intervals and then converted to the net physical displacement of the top surface of the ball. A threshold was calculated individually for each experiment (1–3 cm/s), depending on the noise levels of the mouse tracker, and if the average speed of each trial fell above the threshold, the mouse was said to be running in that trial. Running speed of the animal was used to divide trials into running and still states that were analyzed separately. Data acquisition was performed using an Intan Technologies RHD2000-Series Amplifier Evaluation System, sampled at 20 kHz; recording was triggered by a TTL pulse at the moment visual stimulation began. Spike responses during a 1000 ms period beginning 500 ms after stimulus onset were used for analysis. LFP power was computed with and without visual stimulation, and each was compared to its power during a time window of similar duration preceding optogenetic stimulation. Raw data collected at

20 kHz were first bandpass filtered between 1 and 300 Hz, and then wavelet transform was used to compute its spectrum (*Torrence and Compo, 1998*). Finally, power spectra were adjusted for $1/f^\alpha$ with $\alpha$ of 2.5 and averaged over all trials for each recording channel.

Cortical layers were estimated by performing current source density (CSD) analysis on data collected during presentations of a contrast-reversing square checkerboard (0.04 cycles/degree, square-wave reversing at 1 Hz). Raw data filtered as above were averaged across all 1 s positive-phase presentations of the checkerboard. Data from channels at the same depth were averaged together within a shank of the electrode, then CSD for each channel was computed as described (*Mitzdorf, 1985*) (see *Figure 2—figure supplement 2*).

## Single-neuron analysis

The data acquired using 128-site microelectrodes were sorted using MountainSort (*Chung et al., 2017*), which allows for fully automated spike sorting and runs at 2× real time. Manual curation after a run on 40 min of data takes an additional 20 min, typically yielding 90 (range 50–130) isolated SUs. Using average waveforms of isolated SUs recorded from V1, three parameters were defined in order to classify SUs into NS or BS (*Niell and Stryker, 2008*). The parameters were as follows: the height of the positive peak relative to the negative trough, the slope of the waveform 0.5 ms after the negative trough, and the time from the negative trough to the peak (see *Figure 2J*). For dLGN recordings, STAs were used to classify units into SUs and MUs (*Figure 7A*).

## Mutual information

Neuronal responses are considered informative if they are unexpected. For example, in the context of visually evoked neural activity, if a neuron responds strongly to only a very specific stimulus, for example, photographs of Jennifer Aniston (*Quiroga et al., 2005*), the response is informative. In contrast, if a neuron consistently produces similar responses (measured in the number of spikes per second) to all stimuli that are presented, its responses provide little information. This notion can be formalized by a measure of information called the Shannon entropy,

$$H(X) = E_X[I(x)] = -\sum_{x \in X} p(x).log_2 p(x)$$

where H(X) is in units of bits. The neuron that responds to Jennifer Aniston's face has high entropy and is therefore said to be informative. The concept is further extended to mutual information, $I(R, S)$, which quantifies how much information the neuronal response (R) carries about the visual stimulus (S) by computing the average reduction in uncertainty (entropy) about the visual stimulus produced by observing neuronal responses. Intuitively, observing responses from the aforementioned Jennifer Aniston neuron leaves little uncertainty as to which face was presented. Mutual information between S and R is calculated as follows:

$$I(R, S) = H(S) - H(S|R) = E_X[I(x)] = \sum_{r \in R} \sum_{s \in S} p(r, s).log_2 \left( \frac{p(r, s)}{p(r)p(s)} \right)$$

where r and s are particular instances from the set of neural responses (measured as spike counts) and stimuli (grating movement directions in our case), respectively. We used Information Theory Toolbox in MATLAB to compute mutual information (https://www.mathworks.com/matlabcentral/fileexchange/35625-information-theory-toolbox).

## Population-based analysis: decoding the visual stimulus from population responses

To decode the stimulus that the mouse was viewing from single-trial responses of the population of neurons that was recorded simultaneously, we used a linear discriminant analysis (LDA) classifier (*Dadarlat and Stryker, 2017*). The LDA classifier was trained to classify single-trial neural responses, assuming independence between neurons (a diagonal covariance matrix). The trials were divided into two groups of equal size, a control group with the laser off and an experimental group with the laser on. We then randomly subsampled the trials from each group 50 times to obtain a distribution of errors in decoding the stimulus based on the data included.

The difference between the decoding errors using trials from the control and experimental groups was a measure of the effect of optogenetic modulation of nRT-cell responses on the representation of information in V1 about which grating stimulus was viewed. We used a leave-one-out (LDA-LOOXV) approach to train and test classification separately for the trials in each group using MATLAB's fitcdiscr and predict functions. To decode only grating orientation and not movement direction, we grouped stimuli moving 180° apart into the same class.

## Population-based analysis: decoding with equal population spike counts

A higher firing rate has the potential to convey more information. To determine whether the effect of optogenetic manipulation of the nRT on the representation of information in V1 was solely due to the resulting change in V1 firing rates, we compared decoding accuracy from trials in the laser-off and laser-on groups with equal population spike counts, the sum of spikes from all neurons. This was accomplished by selecting subsets of neurons from the population (1–70 neurons were randomly subsampled with replacement). The constructed datasets retain higher-order structure between neural activity within each trial, allowing us to consider many samples of laser-off and laser-on trials that have the same population spike counts. We used an LDA-LOOXV to train and test classification separately for each subset. For each number of neurons, we subsampled with replacement 100 times from the population, yielding 100 combinations of neurons. Classifiers were trained separately on each subsample and for each condition.

## Immunostaining, retrograde staining, microscopy, and image analysis

Immunohistochemistry on mouse brain sections and image analysis were performed as previously described (*Clemente-Perez et al., 2017*; *Ritter-Makinson et al., 2019*). Briefly, the brains were removed, post-fixed for 24 hr in 4% paraformaldehyde (PFA), and then transferred to 30% w/v sucrose in PBS, at 4°C for cryoprotection. Serial coronal sections (30 μm thick) were cut on a Leica SM2000R Sliding Microtome. Sections were stored in 96-well plates in cryoprotectant solution (0.5 M sodium phosphate buffer, pH 7.4, 30% glycerol, and 30% ethylene glycol) at 4°C until further processing. Free-floating sections were washed in PBS, permeabilized with 0.5% Triton-X100 in PBS (PBST-0.5%), washed with 0.05% Triton-X100 in PBS (PBST-0.05%), and blocked in 10% normal donkey serum in PBST-0.05% for 1 hr at room temperature. Primary incubation was performed in 3% normal donkey serum overnight at 4°C, followed by 3 × 10 min washes in PBST-0.05%. Secondary incubation was performed in 3% normal donkey serum overnight for 1–2 hr at room temperature, followed by washes in PBST-0.05% and PBS. All secondary antibodies were dissolved at 1:1000 dilution in 3% NDS in PBST. Sections were mounted in an antifade medium and imaged using either a Biorevo BZ-9000 Keyence microscope or a Zeiss LSM880 confocal microscope. The expression of the viral constructs in different brain regions was confirmed with reference to two standard mouse brain atlases (*Paxinos and Franklin, 2001* and the Allen Brain Atlas *Lein et al., 2007*).

In the experiments described in *Figure 1* and *Figure 1—figure supplements 1* and *2*, sections were immunostained with antibodies against PV (host: rabbit; concentration: 1:500) and donkey anti-rabbit Alexa Fluor 594 (concentration: 1:500). In the experiments described in *Figure 1—figure supplement 2*: 30 nl of Cholera-ToxinB conjugated with Alexa488 was injected in the dLGN in the adult male mouse at −2.8 mm from Bregma, 2.2 mm lateral from midline, 2.4 mm under the cortical surface using the same injection method as described for opsin injections. 48 hr post-CTB injection, mice underwent cardiac perfusion/fixation with ice-cold PBS followed by 4% w/v PFA in 0.1 M sodium phosphate (PBS) as described above. The primary antibodies used in the immunohistochemical procedures were as follows: polyclonal rabbit anti-somatostatin-14 (SST), 1:1000 (Peninsula Laboratories); rabbit antiserum against PV, 1:500. For dual fluorescent immunostaining (PV/SST), a combination of primary antibodies—monoclonal mouse anti-PV, 1:1000 and polyclonal rabbit anti-SST, 1:1000 (Peninsula Laboratories)—was applied to sections and incubated overnight at 4°C (shaker). All primary antibodies were dissolved in 3% NDS in PBST. After three washes in PBST, monoclonal primary antibodies were visualized by incubation in the dark for 2 hr with appropriate secondary fluorochrome-conjugated antibodies: donkey anti-rabbit Alexa Fluor 549, donkey anti-rabbit 647, and donkey anti-mouse Alexa Fluor 555 (Invitrogen).

## Experimental design and statistical analysis

The experiments reported here were designed to determine (1) whether specific cell types in nRT project to visual thalamus and (2) whether optogenetic activation or inhibition of either SST or PV cells in nRT alters visual responses and oscillatory activity in V1 and dLGN. For the quantification of putative synaptic inputs from SST and PV nRT neurons to various thalamocortical nuclei (see *Figure 1*), we performed immunohistochemistry of brain sections from n = 2 *Sst-Cre* and n = 3 *Pvalb-Cre* mice (male, age range 2–4 months) in which we had injected an AAV construct that resulted in eYFP expression in SST or PV neurons, respectively (for specific details, see *Figure 1*). For the physiology experiments, we recorded V1 responses of n = 4 (two males and two females) *Sst-Cre* mice with optogenetic activation, n = 4 (two males and two females) *Sst-Cre* mice with optogenetic inhibition, n = 4 (three males and one female) *Pvalb-Cre* mice with optogenetic activation, and n = 2 (males) *Pvalb-Cre* mice with optogenetic inhibition (age range 3–5 months). Furthermore, we recorded dLGN responses in n = 3 (males) *Sst-Cre* mice with optogenetic activation and n = 3 (males) *Sst-Cre* mice with optogenetic inhibition. For the control experiment, we recorded n = 2 mice (males, age 3 months), which were injected with a viral construct expressing eYFP in nRT. The expression and specific location of the opsins were verified in all the recorded mice listed above (see representative examples in *Figure 1—figure supplement 1*). All data are illustrated in *Figures 1–8* and figure supplements.

All numerical values are given as mean ± SEM and error bars are SEM unless stated otherwise in the figure legends. Hypothesis testing for representative channels or cells in one mouse was done using appropriate parametric or non-parametric tests. Due to the nested structure of data, a multi-level analysis was used to probe statistical differences between measurements under different conditions across all mice (*Aarts et al., 2014*). Number of samples (n), number of subjects (when applicable), exact p-value, and test name are reported in figure legends and tables. Data analysis was done in MATLAB, Origin 9.0, GraphPad Prism 8, R-project, and SigmaPlot using Wilcoxon rank-sum, Wilcoxon signed-rank test, Spearman rank-order correlation with the Bonferroni correction for multiple comparisons.

## Acknowledgements

JTP is supported by NIH/NINDS grant R01NS096369, Gladstone Institutes, the Kavli Institute for Fundamental, DoD (EP150038), and NSF award #1608236. MSH is supported by NSF award 1822598 and by NIH grant R01EY025174. BH is supported by the American Epilepsy Society Postdoctoral Research Fellowship. MPS is supported by NIH grants R01EY002874 and R01EY025174. MPS is a recipient of the Disney Award for Amblyopia Research from Research to Prevent Blindness. AHC is supported by the Berkelhammer award. FSC is supported by the National Science Foundation Graduate Research Fellowship and the National Research Service Award Fellowship (NRSA, NINDS). ACP is supported by National Science Foundation Graduate Research Fellowship awards #1650113. This work was also supported by an NIH/NCRR grant (C06 RR018928) to Gladstone Institutes. We thank Marie Burkart and Stephanie Holden for assistance with immunohistology, and Meredith Calvert of Gladstone Histology and Light Microscopy Core for help with confocal microscopy. We also thank Kathryn Claiborn for critical feedback on our manuscript.

## Additional information

### Funding

| Funder | Grant reference number | Author |
|---|---|---|
| National Institute of Neurological Disorders and Stroke | R01NS096369 | Jeanne T Paz |
| National Science Foundation | 1822598 | Michael P Stryker |
| National Institute for Health Research | EY025174 | Michael P Stryker |
| American Epilepsy Society | | Bryan Higashikubo |
| National Institute of Neurolo- | F31NA111819 | Frances S Cho |

| | | |
|---|---|---|
| gical Disorders and Stroke | | |
| Gladstone Institutes | | Jeanne T Paz |
| DOD | EP150038 | Jeanne T Paz |
| National Science Foundation | 1608236 | Jeanne T Paz |
| National Institutes of Health | R01EY002874 | Michael P Stryker |
| Research to Prevent Blindness | Disney Award for Amblyopia Research | Michael P Stryker |
| National Center for Research Resources | C06 RR018928 | Michael P Stryker |

The funders had no role in study design, data collection and interpretation, or the decision to submit the work for publication.

### Author contributions

Mahmood S Hoseini, Conceptualization, Data curation, Software, Formal analysis, Validation, Investigation, Visualization, Methodology, Writing - original draft, Writing - review and editing; Bryan Higashikubo, Data curation, Funding acquisition, Validation; Frances S Cho, Validation, Visualization, Methodology, Writing - review and editing; Andrew H Chang, Validation; Alexandra Clemente-Perez, Validation, Investigation; Irene Lew, Validation, Methodology; Agnieszka Ciesielska, Investigation; Michael P Stryker, Conceptualization, Resources, Supervision, Funding acquisition, Writing - original draft, Writing - review and editing; Jeanne T Paz, Conceptualization, Resources, Supervision, Funding acquisition, Validation, Investigation, Methodology, Writing - original draft, Writing - review and editing

### Author ORCIDs

Mahmood S Hoseini https://orcid.org/0000-0002-3139-0561
Frances S Cho https://orcid.org/0000-0002-2576-7801
Irene Lew https://orcid.org/0000-0003-0153-1236
Michael P Stryker https://orcid.org/0000-0003-1546-5831
Jeanne T Paz https://orcid.org/0000-0001-6339-8130

### Ethics

Animal experimentation: We performed all experiments in compliance with protocols approved by the Institutional Animal Care and Use Committees at the University of California, San Francisco and Gladstone Institutes (protocol numbers AN180588-02C and AN174396-03E). Precautions were taken to minimize stress and the number of animals used in all experiments. We followed the NIH guidelines for rigor and reproducibility of the research.

### Decision letter and Author response

Decision letter https://doi.org/10.7554/eLife.61437.sa1
Author response https://doi.org/10.7554/eLife.61437.sa2

## Additional files

### Supplementary files

• Source data 1. Source data spreadsheet.
• Transparent reporting form

### Data availability

All data generated or analyzed during this study are included in the manuscript and supporting files. Source data for all figures are available in a spreadsheet format.

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
