## [Decision Letter]

**Acceptance summary:**

This paper investigates two different cell types within the reticular nucleus of the thalamus (nRT), demonstrating that they have different projection patterns and impacts on subsequent visual processing. The overarching theme here is to understand if and how parvalbumin-expressing and somatostatin-expressing cells in nRT are involved in gating information to the cortex during different behavioral contexts. While some of this manuscript reinforces and complements work by this group and others, the additions of a mutual information analysis and the emphasis on these nRT cell types within the visual system are both novel.

**Decision letter after peer review:**

Thank you for submitting your article "Gamma rhythms and visual information in mouse V1 specifically modulated by somatostatin+ neurons in reticular thalamus" for consideration by *eLife*. Your article has been reviewed by 3 peer reviewers, and the evaluation has been overseen by a Reviewing Editor and Laura Colgin as the Senior Editor. The following individual involved in review of your submission has agreed to reveal their identity: Ashley L Juavinett (Reviewer #1).

The reviewers have discussed the reviews with one another and the Reviewing Editor has drafted this decision to help you prepare a revised submission.

Summary:

The diversity of cell types within the thalamic reticular nucleus (nRT) has been increasingly appreciated as has its influence on cortical responses depending on the behavioral context. Here, the authors show how a subset of nRT neurons expressing somatostatin project into the primary thalamic nucleus of the visual system, the lateral geniculate nucleus, and influence visual processing and rhythm generation in the primary visual cortex. This study highlights the functional effects of a specific circuit within the nRT, supporting the hypothesis that cell-type specific pathways within the nRT subserve distinct functions in sensory processing.

The authors' overarching theme is to understand if and how PV and SOM cells in the thalamic reticular nucleus (nRT) are involved in gating information to the cortex during different behavioral contexts (movement and visual input). The authors provide anatomical data showing that SOM neurons of the visual sector of the nRT project to the lateral geniculate nucleus (LGN). Using optogenetics to show loss and gain of function influences, the authors demonstrate that SOM neurons (and not PV) of the nRT disrupt thalamocortical transmission and gamma activity in the primary visual cortex (V1) and LGN, thus implicating a cell-type specific pathway from the nRT in thalamocortical modulation of visual information processing and gamma oscillations.

Overall, the reviewers agreed that the manuscript presents an interesting set of data but would benefit from removing claims not well supported by the presented data, more clearly motivating the experiments and analyses performed, and clarifying several claims and aspects of the methods including the statistical treatment of certain results.

Essential revisions:

Claims not well supported by the presented data:

1. The discussion of V1 layer specificity is not supported by the data. On page 6, column 2, line 16: "Visual stimuli evoked the strongest responses in the superficial layers (~layer 2/3) and layer 4 (Figure 2 – S3A, B)." This result is not quantified nor is it evident in the figures. Relatedly, "optogenetic activation of SOM nRT neurons reduced the activity evoked by visual stimuli in these layers (Figure 2 – S3C)." This manipulation reduced the activity in all layers. Comparisons between layers were not formally performed and responses of the shanks shown in S3E differ in their profile. Considering these factors, the appropriate analyses should be performed or the text and related title of the supplemental figure, "Optogenetic activation of SOM neurons in nRT has a stronger effect on neuronal responses in superficial layers in V1," should be changed.

2. While the greatest change was seen in gamma during running, it appears optogenetic activation of SOM neurons in the nRT reduced power across all frequencies whether the mouse was still or running (Figure 3). The authors assert that inhibiting SOM neurons produced a "consistent increase" in both gamma and spiking activity of cells in V1. However, the stats in Figure 5A do not support a significant increase in beta or gamma. Thus, it is unclear why one of the overarching conclusions of the paper is the selective impact on gamma activity.

3. Page 9, line 17: "presumably because of their strong projections to the dLGN" – why not any of the other changes in V1 already described?

4. "These findings suggest that SOM neurons in nRT can powerfully modulate the encoding ability of neurons in dLGN" – it is unclear how the CV captures encoding ability. Similarly, on page 10, line 15: "determines how accurately visual stimuli are encoded" and page 12 "positioned to control encoding ability" – the authors should be careful about claims about accurate encoding vs the ability to perform an accurate decoding.

5. Effects on LGN activity. The stats presented in Figure 8A indicate that the bidirectional changes in power were not significant. However, the explanation in the text is confusing and also fails to take these stats into account.

6. Page 12, line 27: "and which is associated with visual attention," – nothing in this paper elucidates mechanisms of visual attention and this mention should be removed. Similarly, line 40, "leading to a blurred representation of the visual world" is an interesting possible extension of these findings but is not directly supported by the data. Whether the visual world is "blurred" or whether stimuli are not easily decoded from neural activity are not equivalent.

Results requiring further clarification:

1. It would be worthwhile elaborating on the relationships between PV and SOM neurons in the visual sector of nRT. The results in Figure 1 – S1 should be quantified rather than simply showing representative images. Clemente-Perez et al. 2017 does not provide clear evidence that the two groups are mutually exclusive, but rather shows co-expression of PV and SOM in some nRT neurons. The relationship between the labeled neurons in the two Cre-lines should also be clarified. The results from Martinez-Garcia et al. 2020 and Li et al. 2020 should also be integrated into the manuscript.

2. Furthermore, there appears to be a substantial population of PV neurons in the visual sector, but it is unclear if these are projection neurons. If so, where do they terminate since visual thalamic nuclei are largely devoid of PV terminals? If they are intrinsic to the nRT, optogenetic stimulation of PV could potentially silence SOM projection neurons and disrupt thalamocortical activity through the LGN. Does optogenetic activation of PV affect LGN activity? How these factors affect the interpretation of the results should be addressed.

3. Another, more fundamental, issue with these experiments is the location chosen for injection. Retrograde labelling experiments have suggested that dLGN projecting nRT neurons are located more posterior and dorsal (approx. A/P: -1.6, M/L: +/- 2.0, D/V: -2.4) than the injection site used here. This raises the concern that PV neurons outside the nRT may be labeled. Moreover, there is no guarantee that PV and SOM positive populations projecting to LGN are located in the same region of nRT. To clearly differentiate the relevance of PV and SOM nRT neurons for visual processing, retrograde labelling of each population would be important to assess the location that should be targeted. Alternatively, the framing of these issues should be modified. Because the reasoning for comparing PV and SOM populations is not particularly clear, it may be better to de-emphasize this set of results and focus on the impact of activating the SOM population on sensory processing.

4. For V1 (and LGN) activity, is it possible that a reduction in spike firing can explain the loss of power? What is the relationship between these variables? This issue also comes up in the Discussion section, and it would be important to articulate whether the reduction in spike firing is tied to the loss of power. If these variables are tightly coupled, then the loss of power could simply be a consequence of reduced activity.

Methods requiring clarification:

1. The statistical test run for each comparison throughout the paper needs to be clearly included. Although the reporting methods form says, " In the main text statistical reporting always directly follows the sentence in which the result is first mentioned, and includes test name, test statistics, exact p-value (when possible)and sample size (n)," this is not the case in the text. Similarly, the reporting form says, " A table has been uploaded with all statistical tests done for figures 2 to 4" but this table does not include Figure 3, and there are many other statistical tests performed throughout the manuscript that should be included. Table 1, and any additional tables that are added, should also include the number of observations for each comparison.

2. Furthermore, the manuscript contains statements and conclusions about the magnitude of change. In many instances the authors use inferential statistics to illustrate the magnitude of effects (e.g. "moderate decreases in theta and beta and strong decrease in gamma"). If the magnitude of change is important, the authors should consider using estimation statistics. As it stands, the p-valued based stats do not support qualitative statements about the magnitude of change.

3. In some figures, it is difficult to identify what each dot represents, for example, in Figure 9D. It would be useful to specify n values for all figures (not just those included in Table 1) and identify the quantity (i.e. n = X neurons or n = Y pairs) to help the reader to understand what is being shown.

4. There is no description in the methods about the acquisition of LFP band and calculation of power, 1/f scaling, etc. Was the LFP band acquired at 20 Hz as well? Was it down-sampled? How was it filtered? Relatedly, is Figure 2L showing power across all bands, and how exactly was the CSD / laminar structure computed?

5. Information theory and the concept of mutual information are not fully introduced in the main text. Similarly, the motivation and caveats of the LDA and deep neural network analyses should be more explicit. Relatedly, it is important to frame the methods description of the mutual information analysis first within information theory (and with the proper citation), because "Neuronal responses are considered informative if they are unexpected," does not make sense taken out of that specific context.

Organization of the manuscript, including motivation for some experiments:

1. It would be helpful to readers to more clearly define the experimental question at the outset and center the organization and flow of the manuscript accordingly. Whereas the focus of the introduction is largely on the role of gamma in visual attention and perception, these concepts are not central to the paper and are never directly tested. Other ideas dominate the current figures: the role of narrow- and broad-spiking cells and the differences in neural activity in specific cell types between moving and stationary. Framing these questions for the reader at the beginning, as well as re-considering the organization of some of the figures and supplementary information, would be very useful. With 9 figures and many supplemental figures, the main message in this paper is easily obscured, and the manuscript would benefit from a tighter framing of this interesting collection of data.

2. One reason that the different results presented sometimes appear disconnected is that the writing is at times disjointed and lacks transitions. Even in the abstract (abstract lines 14-16) following a sentence describing projections of nRT neurons, the authors immediately jump to a describe "powerful modulation" of sensory encoding and gamma activity without stating what this modulation is or even mentioning that it is produced by SOM-positive nRT neurons. Similar examples occur for many key transitions in different parts of the manuscript (For example: Intro., p. 2, line 1-2, left; Intro., p. 2, line 33-38, left; Results, p. 6, line 1-3, left; Results, p. 7, line 4-12, right; Results, p. 10, line 4-8, right). It would be very important to clarify the reasoning for each of these steps individually and together, so that reader can understand the results.

3. Another example needing clarification is the transition from comparing the impact of SOM and PV manipulations on gamma oscillations to examining the impact on sensory responses in V1 using mutual information (MI) and linear decoding methods. This is a large jump and seems almost the beginning of a separate study. More to the point, the authors do not seem to attempt to relate the observed effects on oscillatory dynamics to their various analysis of sensory encoding in the dorsal lateral geniculate nucleus (dLGN) and visual cortex (V1) beyond a general discussion. To address this issue, it would be important to clearly link changes in sensory encoding and processing with the observed changes in oscillatory dynamics and, ideally, to better describe how optogenetic manipulation of nRT leads to changes in each aspect of sensory processing. Unifying the various observations in this way would substantially strengthen the study and would allow the authors to address their original thesis that gamma oscillations are relevant to sensory processing and that the role of the nRT in "gating" sensory input involves control of these oscillatory dynamics.

4. The authors present multiple analysis, including linear decoding, spike triggered averages (STA), measurement of coefficients of variation (CV) and a pairwise correlation based decoding method without discussing how effects observed in one measure relate to other observations.

5. Overall, the Discussion section could be improved with a more in-depth discussion about the following: identity, source, and targeting of SOM and PV neurons in the visual of nRT, the relationship between power and levels of activity and candidate circuits and their degree of independence in modulating V1 activity.

6. As touched on in the Discussion, it is not clear how locomotion via VIP neurons and SOM nRT activation contributes independently to activity in V1. Could it be possible that SOM nRT via LGN could regulate VIP neurons in V1? What is the evidence that describes the inputs and outputs of VIP interneurons?

7. The notion that mouse nRT may be suited for switching attention between different sensory modalities is interesting, and the work of the Halassa lab may be worth citing and discussing in the Discussion.

8. It may be worth discussing the relationship with the perigeniculate nucleus as the visual sector of nRT in carnivores. Is the more caudal part of the carnivore nRT analogous/homologous to the auditory and somatosensory sector of mouse nRT?

9. The Discussion would benefit from references to specific figures.

[Editors' note: further revisions were suggested prior to acceptance, as described below.]

Thank you for submitting your article "Gamma rhythms and visual information in mouse V1 specifically modulated by somatostatin+ neurons in reticular thalamus" for consideration by *eLife*. Your article has been reviewed by 3 peer reviewers, and the evaluation has been overseen by a Reviewing Editor and Laura Colgin as the Senior Editor. The following individual involved in review of your submission has agreed to reveal their identity: Ashley L Juavinett (Reviewer #1).

The reviewers all agreed that the manuscript was substantially improved in response to the reviewers' comments. However, after consultation among the reviewers, the reviewers agreed that there were six remaining issues to address before publication in *eLife*.

1. The authors should clarify whether optogenetic inhibition of SST neurons significantly alters gamma power relative to baseline. It seems that inhibiting SST nRT neurons did not significantly increase either gamma or spiking activity relative to baseline, and this could be more clearly acknowledged. Furthermore, there at times seems to be contradictions between the main text and the figures as well as with the response to the reviewers. These contradictions should be resolved in a revised manuscript.

2. The authors could report the P values based on the adjusted alpha (0.05/number of comparisons) for Bonferroni corrections.

3. The heading "Optogenetic activation of SST nRT but not PV nRT neurons diminished encoding ability of BS cells in V1" should be changed to "Optogenetic activation of SST nRT but not PV nRT neurons diminishes encoding ability of both BS and NS cells in V1" as both BS and NS encoding were affected.

4. The authors should further clarify the extent of PV and SST co-expression (see points 4a-4f).

4a. In Figure 1- supplemental figure 1, some images appear to show that PV and SST are mutually exclusive while the CTB experiments show some co-expression. These latter retrograde tracing experiments indicate that PV, SST and co-expressing neurons project to LGN.

4b. The authors should clarify why in Figure 1-supplemental figure 1, there does not appear to be co-localization between ChR2-EYFP and PV IHC in PV-Cre;ChR2-EYFP mice.

4c. The possibility of co-expression should be incorporated into the interpretation of the results.

4d. The possibility for interactions between PV and SST neurons in nRT should be incorporated into the authors' interpretation of their results.

4e. The reference to Campbell et al. 2020 regarding PV and SST expression in the visual sector of nRT should be eliminated as their comparison was between SST and NeuN.

4f. The figure legend in Figure 1C, D should be changed to SST for consistency. Consider changing SOM to SST throughout (ie. Figure 1—figure supplement 2).

5. It is difficult to interpret the effects of activation of SST-positive nRT neurons on population decoding and mutual information measures. These effects could simply be due to non-specific inhibition changes produced by activating a random subset of this population introducing an additional source of noise to the sensory signal. While the observation that most single-units recorded in LGN were inhibited may reduce this concern, it is unclear whether this is sufficient to address the concern. A more useful approach would be to compare sensory response properties of LGN neurons with and without SST activation. This could be done qualitatively by comparing receptive field properties of LGN neurons and the coefficient of variation (which was removed from the previous revision) could be used to support the claim that changes in response fidelity were present at the level of individual neurons and thus that the changes in population decoding were not a consequence of inhomogeneous. If the authors choose not to include such an analysis, this caveat should be included explicitly in the discussion.

6. There remains some lack of clarity in the text. Some claims are very difficult to understand from what is written (see, for example, the case mentioned in Concern 1). The complexity of some claims also makes it difficult for the authors to draw specific conclusions in many cases. For instance, after discussing a variety of changes in oscillatory dynamics across several frequency bands during locomotion and non-locomotion produced by bi-directional optogenetic manipulations of nRT sub-populations the authors conclude that "SST nRT neurons are well positioned to exert powerful effects on the visual input to V1". Moreover, the relationship between oscillatory dynamics and sensory responses of individual neurons is still somewhat unclear. These challenges should be mitigated by a careful discussion of the relationship between changes in oscillatory dynamics, firing rates and the relevant neuronal populations in the thalamus and cortex and further clarification of the language reporting the results in a revised manuscript.

Reviewer #1:

In this paper, the authors have dissected two different circuits arising from somatostatin (SST) and parvalbumin (PV) positive cells from the nRT nucleus of the thalamus. This paper is an impressive technical tour de force, incorporating many techniques to address the underlying questions about the connectivity of these cells and their role in visual processing. With this lab's overarching interest in brain oscillations, there is a particular focus on the impact of these cells on gamma in the visual cortex. This work fits into a broader body of work that is continuously seeking to elucidate the distinct roles of various cell types in particular areas of the brain, and provides possible mechanistic explanations for previous findings from other labs.

The strengths of this paper are largely derived from the comprehensive approach to the overarching question. First, the authors show that SST and PV neurons in the nRT have very different projection patterns to other nuclei in the thalamus, both visual and somatosensory. Seeing that SST neurons project more strongly to other visual nuclei (such as the dLGN), they hypothesized that optogenetically manipulating the activity of these neurons would more strongly impact the visual cortex (which primarily receives input from the dLGN). While doing so impacted multiple frequency bands across multiple layers of cortex, the impact on gamma was most evident. Importantly, the authors performed control experiments for non-specific laser effects. Although the analysis on broad spiking (BS) and narrow spiking (NS) cells did not yield any differences, future studies looking at more specific cell types may.

As V1 is ultimately responsible for encoding visual information, the authors then tested whether or not SST activation would reduce mutual information between responses in broad spiking neurons in V1 – and it does. Additional controls demonstrated that this was not due to spike rate differences. Future work could investigate whether this is behaviorally relevant to the animal during a discrimation task, for example.

While gamma oscillations are strongly correlated with visual attention, this isn't directly tested in this paper. The authors have provided a potential mechanism by which gamma could be regulated, but have not directly confirmed whether this is the mechanism that underlies changes during attention. Future work would need to address this directly using a behavioral task that probes visual attention. Overall, the authors have addressed the feedback on the initial submission and the claims made in this paper are supported by the data.

The authors have comprehensively addressed the feedback of the first submission, and have delivered a much stronger manuscript as a result. I am particularly impressed by the new statistical approach and the additional sections in the methods. As such, I have no additional major suggestions. There are some thoughts on additional behavioral experiments, which I'm sure will come of no surprise to the authors.

I have two remaining concerns about the statistical approach:

1. Why not use the nested approach in Figure 6?

2. There is a mention of Bonferroni corrections in the methods but the reported P-values are still * p < 0.05. Use of a Bonferroni correction means adjusting the alpha – reporting the adjusted alpha (0.05/number of comparisons) would make sense here.

The authors might reconsider the opening of the paper, which largely focuses on gamma oscillations. Although this is indeed one of the main findings of the manuscript, it is not the only finding, and in this reviewer's perspective, this manuscript offers quite a bit by way of understanding the cell types within a previously unknown thalamic nucleus, and also in the mutual information approach. Regardless, the logical progression of the introduction is clear and I can appreciate the lab's prevailing interest in gamma oscillations and their role in visual attention.

One small suggested change: the header "Optogenetic activation of SST nRT but not PV nRT neurons diminishes encoding ability of BS cells in V1" is a bit misleading, because it impacts both BS and NS cells. I suggest, "Optogenetic activation of SST nRT but not PV nRT neurons diminishes encoding ability of both BS and NS cells in V1."

Lastly, the authors have provided "source data" but this is simply a word document of the two tables in the manuscript. I do not see a spreadsheet of source data as is referenced in "Data Availability" statement on the cover sheet. The authors might consider making their data and relevant code available.

Reviewer #2:

Many of the concerns raised by the reviewers have been adequately addressed. However, there is still confusion about the exclusivity of PV and SOM neurons in nRT. It is also not clear how such heterogeneity or lack thereof, impact the results and interpretation. The authors state that the two groups are mutually exclusive but that a subgroup "show both markers". Which is it and how does this impact the interpretation of results? The bottom line based on the authors results is to acknowledge anatomically there is sparse co-expression in the visual sector, but functionally SST projecting neurons seem to regulate dLGN activity related to gamma.

Other related issues are listed below.

The reference to Campbell et al. 2020 is inaccurate, since in this paper they showed about 80% of neurons in the visual sector co-express SST and NeuN (not SST and PV). There are several instances throughout the paper that mistakenly cite Campbell et al. 2020 as evidence for PV and SST co-expression. This needs to be amended.

In the supplemental figure 1, the authors showed with IHC, the labeling of PV in SST-Cre ChR2-EYFP mice. Based on the high-power images it appears that SST and PV are mutually exclusive in the visual sector of nRT (head), but it doesn't appear this was quantified; was there any evidence of co-expression (as asked by one of the reviewers) in the visual sector? CTB experiments seem to reveal some co-expression but from the images it's difficult to evaluate how prevalent.

In the supplemental figure 1, I am puzzled by the lack of co-expression between PV-Cre/ChR2 eYFP and PV IHC. Do the authors have an explanation?

While the authors show that PV projections largely project to PV and not dLGN, they still failed to address whether intrinsic connections between PV and SOM in nRT could complicate their optogenetic manipulations.

Figure 1 C-D, the bar graphs refer to SOM, for consistency it should be changed to SST (see also figure 2 supplement panels C-E).

Reviewer #3:

The author's study brings together a set of interesting experiments designed to investigate the role of the inhibitory nucleus reticularis of the thalamus (nRT) in controlling sensory processing in the visual cortex and relate this control to changes in gamma oscillatory dynamics that have been tied to attentional control and state-specific sensory processing. By specifically controlling molecularly defined groups of nRT neurons using optogenetic techniques, they find that somatostatin- (SST-) positive neurons play a distinct role in regulating visual cortical oscillations as well as population encoding across multiple conditions. Although this is a valuable discovery, there are some conceptual questions that remain unanswered. First, it is unclear how to interpret the changes in population decoding observed in V1 following SOM neuron activation. Because the optogenetic approach used would randomly activate a subset of SST neurons, the change in the representation they estimate may simply represent an injection of noise into the signal arising from the non-specific inhibitory input. While their linear decoding approach suggests that the effect is not simply due to reduced spiking, the issue of non-specific inhibition is difficult to exclude based on the findings presented. In addition, while their optogenetic experiments convincingly show that activation of SST-positive nRT neurons has a much stronger effect on cortical activity compared with similar manipulation of parvalbumin- (PV-) positive nRT neurons, the anatomical tracing results they presented leave some uncertainty as to whether this population is the main inhibitory projection from the visual subdivision of the nRT to the primary visual thalamus (the lateral geniculate nucleus, LGN). In particular, without quantification the retrograde tracing shown cannot exclude the possibility that PV neurons also substantially project to the LGN. Finally, although the authors do show that both gamma oscillations and population encoding are impacted by changes in SST activity, how these changes related to each other remains unclear. Despite these remaining questions, however, this study does represent valuable progress towards understanding how inhibitory control systems in the thalamus impact key aspects of cortical processing.

While valuable results are presented in the manuscript some technical and conceptual issues remain that complicate the interpretation of these results. Although revisions made to the previous submission have improved the manuscript, particularly by appropriately weakening and/or clarifying some claims, some major concerns still present issues for the overall manuscript which it would be important to address. These concerns are described in detail below:

1. In the previous submission concerns were raised regarding the claim that the majority of neurons projecting from the visual nRT to the LGN are SST-positive, with PV-positive neurons projecting instead to higher-order visual thalamic neurons (Previous Concern 7 and 8) and that the injection site used for the retrograde labelling may not have fully addressed these concerns (Previous concern 9). To address these concerns, the authors performed retrograde labelling experiments using cholera-toxin (CTB-Alexa488) injected into the LGN followed by histological assessment of labelled neurons to determine whether they were PV- or SST- positive. Based on these experiments, it appears that both cell types project to LGN. However, based on their earlier viral tracing experiments, they still conclude that the predominant projection is from SST-positive neurons. While the earlier results do provide qualitative support for this conclusion, it would still be important to have some quantification in order to make the claim that input is "mainly" from SST.

2. Previously, a concern was raised regarding the statistical significance of changes in gamma power produced by suppression of SST neurons (Previous concern 2). This remains an important claim for the overall message but, while the authors rebuttal appears to state that additional statistical analysis has clarified this point, this is not at all clear from the text. The rebuttal response refers to Figure 5A which is described in the main text as follows: "We found that inhibiting eNpHR-expressing SST nRT neurons at baseline produced a consistent, but insignificant, increase in both gamma and spiking activity of cells in V1 and a reduction in theta and beta (Figure 5A, Table 2, Figure 5—figure supplement 1, Figure 2-source data 1).…". Based on this, it is very hard to determine whether any significant change occurred (indeed, the change is described as "insignificant"). Moreover, the analysis in the figure itself appears to be comparing inhibition of PV versus SST rather than making any comparison with baseline although it is difficult to be certain due to some overall issues with labelling (see major concern 4). This key point needs substantial clarification in the main text and legend.

3. It is difficult to interpret the effects of activation of SST-positive nRT neurons on population decoding and mutual information measures these effects could simply be due to the fact that non-specific inhibition changes produced by activating a random subset of this population introduce an additional source of noise to the sensory signal. While the observation that most single-units recorded in LGN were inhibited may reduce this concern, it is unclear whether this is sufficient to address the concern. A more useful approach would be to compare sensory response properties of LGN neurons with and without SST activation. This could be done qualitatively by comparing receptive field properties of LGN neurons and the coefficient of variation (which was removed from the previous revision) could be used to support the claim that changes in response fidelity were present at the level of individual neurons and thus that the changes in population decoding were not a consequence of inhomogeneous inhibitory input. In any case, however, it would be worth including this caveat in the discussion.

4. Despite some improvement compared to the previous submission, there is still substantial lack of clarity in the text. Some claims are very difficult to understand from what is written the text (see, for example, the case mentioned in Major Concern 2). The complexity of some claims also appears to make it difficult for the authors to draw specific conclusions in many cases. For instance, after discussing a variety of changes in oscillatory dynamics across several frequency bands during locomotion and non-locomotion produced by bi-directional optogenetic manipulations if nRT sub-populations the authors conclude that "SST nRT neurons are well position to exert powerful effects on the visual input to V1". Moreover, as before, the relationship between oscillatory dynamics and sensory responses of individual neurons is still somewhat unclear. While these challenges could have been mitigated by a careful discussion of the relationship between changes in oscillatory dynamics, firing rates and the relevant neuronal populations in the thalamus and cortex, such a discussion does not seem to have been included in the current manuscript.

---

## [Author Response]

Essential revisions:Claims not well supported by the presented data:1. The discussion of V1 layer specificity is not supported by the data. On page 6, column 2, line 16: "Visual stimuli evoked the strongest responses in the superficial layers (~layer 2/3) and layer 4 (Figure 2 – S3A, B)." This result is not quantified nor is it evident in the figures. Relatedly, "optogenetic activation of SOM nRT neurons reduced the activity evoked by visual stimuli in these layers (Figure 2 – S3C)." This manipulation reduced the activity in all layers. Comparisons between layers were not formally performed and responses of the shanks shown in S3E differ in their profile. Considering these factors, the appropriate analyses should be performed or the text and related title of the supplemental figure, "Optogenetic activation of SOM neurons in nRT has a stronger effect on neuronal responses in superficial layers in V1," should be changed.

We agree with the reviewer that the manipulation changed the activity in all layers. Therefore, we removed the claim of laminar differences in the text and the Figure legend.

2. While the greatest change was seen in gamma during running, it appears optogenetic activation of SOM neurons in the nRT reduced power across all frequencies whether the mouse was still or running (Figure 3). The authors assert that inhibiting SOM neurons produced a "consistent increase" in both gamma and spiking activity of cells in V1. However, the stats in Figure 5A do not support a significant increase in beta or gamma. Thus, it is unclear why one of the overarching conclusions of the paper is the selective impact on gamma activity.

Figure 3 shows that optogenetic activation of SOM neurons in the nRT reduced power *mainly* in the gamma range. Also, we have now calculated confidence intervals to show that the strongest change occurred in gamma power. We have clarified this in the text (page 4).

We realized that the nested statistic test (Aarts et al., 2014) is more appropriate for analyzing our dataset. We added a paragraph in the “Experimental Design and Statistical Analysis” section in the Materials and methods to explain this. According to the new statistical analysis, the results in Figure 5A show a significant increase in the gamma power, but not in the other power bands. We have now clarified this in the text and Figure legends (page 6).

3. Page 9, line 17: "presumably because of their strong projections to the dLGN" – why not any of the other changes in V1 already described?

We apologize for lack of clarity. Other changes in V1 can also be explained by nRT SOM projections to the dLGN. We have revised to remove the quoted text to address this concern.

4. "These findings suggest that SOM neurons in nRT can powerfully modulate the encoding ability of neurons in dLGN" – it is unclear how the CV captures encoding ability. Similarly, on page 10, line 15: "determines how accurately visual stimuli are encoded" and page 12 "positioned to control encoding ability" – the authors should be careful about claims about accurate encoding vs the ability to perform an accurate decoding.

We revised the text to address this concern. Variability of responses does limit the encoding ability of sensory neurons. Our analysis of the effect of optogenetic modulation of nRT on dLGN responses has two parts. First, we merely considered one neuron at a time, and computed the variability of its responses to identical stimuli, as indicated by the coefficient of variation (CV). To measure the ability of dLGN responses to sustain decoding of the orientation of the grating stimuli used to study V1, we devised a novel method based on the responses of pairs of dLGN neurons. However, the reviewers have a reasonable point. There is no standard way to measure the decoding of information in the activity of neurons in the dLGN. Any such approach is, of course, subject to criticism. Therefore, we decided to remove this part of the results and the associated parts in the methods section.

5. Effects on LGN activity. The stats presented in Figure 8A indicate that the bidirectional changes in power were not significant. However, the explanation in the text is confusing and also fails to take these stats into account.

We apologize for this inconsistency. We redid the statistical analysis using the nested statistical approach, which is more appropriate for our dataset (see our answer above to comment #2). In Figure 8A, the new analysis shows that indeed the effects of optical stimulation of ChR2- and eNpHR-expressing SOM nRT neurons produce opposite changes in all the frequency bands except in the theta band during running. Please note that the strongest magnitude of difference between ChR2 and eNpHR effects is in the gamma band (confidence intervals for gamma do not overlap with those for theta and beta comparing eNpHR and ChR2 conditions). We revised the figure by adding the new statistical analysis and edited the legend accordingly (page 16).

6. Page 12, line 27: "and which is associated with visual attention," – nothing in this paper elucidates mechanisms of visual attention and this mention should be removed. Similarly, line 40, "leading to a blurred representation of the visual world" is an interesting possible extension of these findings but is not directly supported by the data. Whether the visual world is "blurred" or whether stimuli are not easily decoded from neural activity are not equivalent.

While we have not measured perception behaviorally, the results are consistent with visual attention behavior. We clarified in the text that our study did not assess visual attention directly, but that our findings that the SOM-nRT->dLGN->V1 pathway alters gamma activity suggest a role for the pathway in visual attention because other findings have associated gamma power with attention (Gray et al., 1992; Singer and Gray, 1995; Kreiter and Singer, 1996).

We agree that the wording about SOM nRT neurons leading to a “blurred representation of the visual world” is too speculative for the Results section, so we deleted this wording.

Results requiring further clarification:1. It would be worthwhile elaborating on the relationships between PV and SOM neurons in the visual sector of nRT. The results in Figure 1 – S1 should be quantified rather than simply showing representative images. Clemente-Perez et al. 2017 does not provide clear evidence that the two groups are mutually exclusive, but rather shows co-expression of PV and SOM in some nRT neurons. The relationship between the labeled neurons in the two Cre-lines should also be clarified. The results from Martinez-Garcia et al. 2020 and Li et al. 2020 should also be integrated into the manuscript.

Thank you for the constructive suggestions. We now elaborate on the relationships between PV and SOM neurons in the nRT (page 3, Results section), and added a new Discussion section on cellular heterogeneity in the nRT (page XX). The results of Figure S1 show the presence of mutually exclusive PV and SOM neurons in the visual sector of the nRT, confirming the findings published in Clemente-Perez et al., 2017, which showed that the PV and SOM neurons in certain sectors of the nRT are exclusive but that a subgroup of neurons shows both markers (see Figure 1 in Clemente-Perez et al., 2017). The two recent manuscripts (Li et al., 2020; Martinez-garcia et al., 2020) which both confirm the cellular heterogeneity in the nRT, are now cited in the manuscript. We also cite Campbell et al., 2020, which showed that the same nRT cells in the visual sector are both SOM and PV positive, although please note that these studies were based on transgenic knockin mice in which transient expression of PV and SOM during early development would cause labeling of the same cell population. In contrast, we used viral labelling, which does not affect precursor cells during development, and which shows the existence of distinct PV and SOM populations of cells in the visual nRT. Not only these cells are distinct, but they also project to distinct visual thalamic nuclei: SOM nRT neurons mainly project to the primary order visual nucleus dLGN, whereas PV nRT neurons mainly project to the higher order visual nucleus LP. Our result cautions against the use of certain transgenic lines if one wants to distinguish these distinct cell populations in the mouse. We have clarified this in the text.

2. Furthermore, there appears to be a substantial population of PV neurons in the visual sector, but it is unclear if these are projection neurons. If so, where do they terminate since visual thalamic nuclei are largely devoid of PV terminals? If they are intrinsic to the nRT, optogenetic stimulation of PV could potentially silence SOM projection neurons and disrupt thalamocortical activity through the LGN. Does optogenetic activation of PV affect LGN activity? How these factors affect the interpretation of the results should be addressed.

There is indeed a population of PV neurons in the visual sector (see Figure 1—figure supplement 1). Please note that all neurons in the nRT (comprising PV and SOM neurons) are projection neurons, meaning that 100% of nRT cells project to the thalamocortical relay nuclei (Pinault et al., 1995; Pinault and Deschênes, 1998). However, we cannot exclude the existence of intra-nRT connections between the projection neurons. Indeed, our Clemente-Perez et al. 2017 study, as well as other studies (Makinson et al., 2017) suggested the existence of such connections. Notably, in Figure 1 and Figure 1—figure supplement 1, we show that PV nRT cells project only weakly to the primary order visual thalamus dLGN but strongly to the higher order visual thalamus LP. We clarified this novel finding in the Results and the Discussion sections (pages 3, 4, 9, and 10).

3. Another, more fundamental, issue with these experiments is the location chosen for injection. Retrograde labelling experiments have suggested that dLGN projecting nRT neurons are located more posterior and dorsal (approx. A/P: -1.6, M/L: +/- 2.0, D/V: -2.4) than the injection site used here. This raises the concern that PV neurons outside the nRT may be labeled.

If the reviewers are concerned that our injections caused opsin expression in PV cells located outside the nRT, this is not the case. Indeed, Figure 1—figure supplement 1 shows that the opsin expression is specific to the nRT. However, we agree that the opsin expression in PV and SOM neurons is in the entire visual sector of the nRT, and not restricted to the visual sector. Nonetheless, although the viral expression may not affect 100% of visual nRT neurons, it has an important advantage compared with transgenic models in that the opsin is only in the nRT (shown in Figure 1—figure supplement 1). Also, please see our reply to comment #7 that points out that PV and SOM cells in mice at the age we studied them are distinct. To address the reviewers’ question as to whether PV and SOM neurons are located in the visual sector of the nRT, we also performed a new experiment: we injected a retrograde tracer (CTB-Alexa) in dLGN and found that SOM but not PV neurons were retrogradely labeled in the visual sector of the nRT. We cannot exclude the existence of PV projections to dLGN, but our results suggest that the projections to dLGN from the visual nRT mainly originate from SOM nRT cells.

Moreover, there is no guarantee that PV and SOM positive populations projecting to LGN are located in the same region of nRT. To clearly differentiate the relevance of PV and SOM nRT neurons for visual processing, retrograde labelling of each population would be important to assess the location that should be targeted. Alternatively, the framing of these issues should be modified. Because the reasoning for comparing PV and SOM populations is not particularly clear, it may be better to de-emphasize this set of results and focus on the impact of activating the SOM population on sensory processing.

Thank you for the suggestion. Based on this comment, we did a new experiment of retrograde labeling of nRT neurons that project to dLGN. Please see our reply to the comment above, which we believe addresses this concern. We now clarify that the main projection from nRT to dLGN originates mainly in SOM nRT neurons, rather than PV neurons. Our results primarily focus on SOM nRT neurons, but we believe that the comparison with PV neurons is very novel and interesting, so we would like to keep these results in the manuscript. However, we edited the text to clarify the location of PV and SOM neurons, and why these would cause distinct effects in V1.

4. For V1 (and LGN) activity, is it possible that a reduction in spike firing can explain the loss of power? What is the relationship between these variables? This issue also comes up in the Discussion section, and it would be important to articulate whether the reduction in spike firing is tied to the loss of power. If these variables are tightly coupled, then the loss of power could simply be a consequence of reduced activity.

We did not find any relationship between these two factors (see Figure 2L). Change in firing rate does not account for much of the variance in gamma power change. We clarify this in the Results section (page 5).

Methods requiring clarification:1. The statistical test run for each comparison throughout the paper needs to be clearly included. Although the reporting methods form says, " In the main text statistical reporting always directly follows the sentence in which the result is first mentioned, and includes test name, test statistics, exact p-value (when possible)and sample size (n)," this is not the case in the text. Similarly, the reporting form says, " A table has been uploaded with all statistical tests done for figures 2 to 4" but this table does not include Figure 3, and there are many other statistical tests performed throughout the manuscript that should be included. Table 1, and any additional tables that are added, should also include the number of observations for each comparison.

Thank you for this important feedback. To address this concern all of the statistical tests have been redone using the more appropriate nested design (see our reply to comment #2). All details are now noted in the legends and the Table.

2. Furthermore, the manuscript contains statements and conclusions about the magnitude of change. In many instances the authors use inferential statistics to illustrate the magnitude of effects (e.g. "moderate decreases in theta and beta and strong decrease in gamma"). If the magnitude of change is important, the authors should consider using estimation statistics. As it stands, the p-valued based stats do not support qualitative statements about the magnitude of change.

We addressed this concern by providing confidence intervals for these changes (see legends of Figures 3, 5, 8).

3. In some figures, it is difficult to identify what each dot represents, for example, in Figure 9D. It would be useful to specify n values for all figures (not just those included in Table 1) and identify the quantity (i.e. n = X neurons or n = Y pairs) to help the reader to understand what is being shown.

Thank you; this has been done.

4. There is no description in the methods about the acquisition of LFP band and calculation of power, 1/f scaling, etc. Was the LFP band acquired at 20 Hz as well? Was it down-sampled? How was it filtered? Relatedly, is Figure 2L showing power across all bands, and how exactly was the CSD / laminar structure computed?

We added this information in the methods section (page 19).

5. Information theory and the concept of mutual information are not fully introduced in the main text. Similarly, the motivation and caveats of the LDA and deep neural network analyses should be more explicit. Relatedly, it is important to frame the methods description of the mutual information analysis first within information theory (and with the proper citation), because "Neuronal responses are considered informative if they are unexpected," does not make sense taken out of that specific context.

We clarified this theory in the Results section (page 6).

Organization of the manuscript, including motivation for some experiments:1. It would be helpful to readers to more clearly define the experimental question at the outset and center the organization and flow of the manuscript accordingly. Whereas the focus of the introduction is largely on the role of gamma in visual attention and perception, these concepts are not central to the paper and are never directly tested. Other ideas dominate the current figures: the role of narrow- and broad-spiking cells and the differences in neural activity in specific cell types between moving and stationary. Framing these questions for the reader at the beginning, as well as re-considering the organization of some of the figures and supplementary information, would be very useful. With 9 figures and many supplemental figures, the main message in this paper is easily obscured, and the manuscript would benefit from a tighter framing of this interesting collection of data.

We have improved the framing of the questions at the end of the introduction section (pages 2 and 3).

2. One reason that the different results presented sometimes appear disconnected is that the writing is at times disjointed and lacks transitions. Even in the abstract (abstract lines 14-16) following a sentence describing projections of nRT neurons, the authors immediately jump to a describe "powerful modulation" of sensory encoding and gamma activity without stating what this modulation is or even mentioning that it is produced by SOM-positive nRT neurons. Similar examples occur for many key transitions in different parts of the manuscript (For example: Intro., p. 2, line 1-2, left; Intro., p. 2, line 33-38, left; Results, p. 6, line 1-3, left; Results, p. 7, line 4-12, right; Results, p. 10, line 4-8, right). It would be very important to clarify the reasoning for each of these steps individually and together, so that reader can understand the results.

Thank you for the constructive suggestion. We revised the abstract and the entire manuscript by adding transitions for clarity as suggested.

3. Another example needing clarification is the transition from comparing the impact of SOM and PV manipulations on gamma oscillations to examining the impact on sensory responses in V1 using mutual information (MI) and linear decoding methods. This is a large jump and seems almost the beginning of a separate study. More to the point, the authors do not seem to attempt to relate the observed effects on oscillatory dynamics to their various analysis of sensory encoding in the dorsal lateral geniculate nucleus (dLGN) and visual cortex (V1) beyond a general discussion. To address this issue, it would be important to clearly link changes in sensory encoding and processing with the observed changes in oscillatory dynamics and, ideally, to better describe how optogenetic manipulation of nRT leads to changes in each aspect of sensory processing. Unifying the various observations in this way would substantially strengthen the study and would allow the authors to address their original thesis that gamma oscillations are relevant to sensory processing and that the role of the nRT in "gating" sensory input involves control of these oscillatory dynamics.

Thank you for this telling comment. The reviewers have raised an important point about the link between the oscillatory dynamics and sensory processing. Elucidating this relationship is beyond the scope of this work, and is properly the subject of a theoretical and computational investigation because it may be understood only in the context of a full model of thalamocortical and cortical function, which does not yet exist.

What we have shown is that oscillatory dynamics and sensory processing, as indicated by firing rates and information, covary, and that both are powerfully modulated by the activity of SOM nRT cells. This finding is an important contribution to the development of such a full model.

We have reorganized the paper and clarified the transitions and figure legends.

4. The authors present multiple analysis, including linear decoding, spike triggered averages (STA), measurement of coefficients of variation (CV) and a pairwise correlation based decoding method without discussing how effects observed in one measure relate to other observations.

We now clarify these relationships in the Results (see our reply to comment #4) (page 6).

5. Overall, the Discussion section could be improved with a more in-depth discussion about the following: identity, source, and targeting of SOM and PV neurons in the visual of nRT, the relationship between power and levels of activity and candidate circuits and their degree of independence in modulating V1 activity.

Thank you for this constructive suggestion. We have added a new section in the Discussion (pages 9 and 10) to clarify these points.

6. As touched on in the Discussion, it is not clear how locomotion via VIP neurons and SOM nRT activation contributes independently to activity in V1. Could it be possible that SOM nRT via LGN could regulate VIP neurons in V1? What is the evidence that describes the inputs and outputs of VIP interneurons?

We thank the reviewer for this comment. The major influences on VIP cells have been described in a reference we now cite (Fu et al., 2014, 2015; Zhang et al., 2014) (page 11).

7. The notion that mouse nRT may be suited for switching attention between different sensory modalities is interesting, and the work of the Halassa lab may be worth citing and discussing in the Discussion.

Done. We have added citations from the Halassa lab that address this point (pages 2 and 9).

8. It may be worth discussing the relationship with the perigeniculate nucleus as the visual sector of nRT in carnivores. Is the more caudal part of the carnivore nRT analogous/homologous to the auditory and somatosensory sector of mouse nRT?

We have elaborated the discussion of the carnivore perigeniculate nucleus in relation to the visual sector of the mouse nRT and our findings in the context of recent findings (page 10).

9. The Discussion would benefit from references to specific figures.

Thank you; this has been done.

References:

Aarts E, Verhage M, Veenvliet J V., Dolan C V., Van Der Sluis S (2014) A solution to dependency: Using multilevel analysis to accommodate nested data. Nat Neurosci 17:491–496.Campbell PW, Govindaiah G, Masterson SP, Bickford ME, Guido W (2020) Synaptic properties of the feedback connections from the thalamic reticular nucleus to the dorsal lateral geniculate nucleus. J Neurophysiol:jn.00757.2019.Clemente-Perez A, Makinson SR, Higashikubo B, Brovarney S, Cho FS, Urry A, Holden SS, Wimer M, Dávid C, Fenno LE, Acsády L, Deisseroth K, Paz JT (2017) Distinct Thalamic Reticular Cell Types Differentially Modulate Normal and Pathological Cortical Rhythms. Cell Rep 19:2130–2142.Fu Y, Kaneko M, Tang Y, Alvarez-Buylla A, Stryker MP (2015) A cortical disinhibitory circuit for enhancing adult plasticity. *eLife* 4:1–12.Fu Y, Tucciarone JM, Espinosa S, Sheng N, Darcy DP, Nicoll RA, Huang ZJ, Stryker MP (2014) A Cortical Circuit for Gain Control by Behavioral State. Cell 156:1139–1152.Gray CM, Engel AK, König P, Singer W (1992) Synchronization of oscillatory neuronal responses in cat striate cortex: Temporal properties. Vis Neurosci 8:337–347.Kreiter A, Singer W (1996) Stimulus-dependent synchronization of neuronal responses in the visual cortex of the awake macaque monkey. J Neurosci 16:2381–2396.Li Y et al. (2020) Distinct subnetworks of the thalamic reticular nucleus. Nature 583:819–824.Makinson CD, Tanaka BS, Sorokin JM, Wong JC, Christian CA, Goldin AL, Escayg A, Huguenard JR (2017) Regulation of Thalamic and Cortical Network Synchrony by Scn8a. Neuron 93:1165-1179.e6.Martinez-garcia RI, Voelcker B, Zaltsman JB, Patrick SL, Stevens TR, Connors BW, Cruikshank SJ (2020) Two dynamically distinct circuits driving inhibition in sensory thalamus. bioRxiv:2020.04.16.044487.Pinault D, Bourassa J, Deschênes M (1995) The Axonal Arborization of Single Thalamic Reticular Neurons in the Somatosensory Thalamus of the Rat. Eur J Neurosci 7:31–40.Pinault D, Deschênes M (1998) Projection and innervation patterns of individual thalamic reticular axons in the thalamus of the adult rat: A three-dimensional, graphic, and morphometric analysis. J Comp Neurol 391:180–203.Singer W, Gray CM (1995) Visual feature integration and the temporal correlation hypothesis. Annu Rev Neurosci 18:555–586.Sokhadze G, Campbell PW, Guido W (2019) Postnatal development of cholinergic input to the thalamic reticular nucleus of the mouse. Eur J Neurosci 49:978–989.Zhang S, Xu M, Kamigaki T, Do JPH, Chang WC, Jenvay S, Miyamichi K, Luo L, Dan Y (2014) Long-range and local circuits for top-down modulation of visual cortex processing. Science (80- ) 345:660–665.

[Editors' note: further revisions were suggested prior to acceptance, as described below.]

The reviewers all agreed that the manuscript was substantially improved in response to the reviewers' comments. However, after consultation among the reviewers, the reviewers agreed that there were six remaining issues to address before publication in eLife.1. The authors should clarify whether optogenetic inhibition of SST neurons significantly alters gamma power relative to baseline. It seems that inhibiting SST nRT neurons did not significantly increase either gamma or spiking activity relative to baseline, and this could be more clearly acknowledged. Furthermore, there at times seems to be contradictions between the main text and the figures as well as with the response to the reviewers. These contradictions should be resolved in a revised manuscript.

We apologize for lack of clarity. Relative to baseline, while optogenetic inhibition of SST nRT neurons significantly increases single-cell firing rates especially during running, the increase in gamma did not reach significance (see Table S1, Figure 5-S1K). We are now clarifying this in the text (page 6, line 182-186).

2. The authors could report the P values based on the adjusted alpha (0.05/number of comparisons) for Bonferroni corrections.

Thank you for the suggestion. This is done. Now each figure caption includes the p value corrected for multiple comparisons.

3. The heading "Optogenetic activation of SST nRT but not PV nRT neurons diminished encoding ability of BS cells in V1" should be changed to "Optogenetic activation of SST nRT but not PV nRT neurons diminishes encoding ability of both BS and NS cells in V1" as both BS and NS encoding were affected.

Thank you for the suggestion. This is done (page 6, line 189).

4. The authors should further clarify the extent of PV and SST co-expression (see points 4a-4f).4a. In Figure 1- supplemental figure 1, some images appear to show that PV and SST are mutually exclusive while the CTB experiments show some co-expression. These latter retrograde tracing experiments indicate that PV, SST and co-expressing neurons project to LGN.

As the reviewer noted, PV and SST neurons defined by cre recombinase expression in adulthood can to some extent express the other marker as previously published (Clemente-Perez et al., 2017) and acknowledged throughout our manuscript (see Introduction page 2, Results page 3-4, and Discussion page 9). What is novel in our manuscript is that perturbing *Pvalb-Cre* and *Sst-Cre* neurons that express ChR2 (through viral injection) has distinct effects on the V1 cortex, which suggests that the effects on V1 are mediated by distinct neuronal outputs. The significance of the findings of this manuscript concern the functional effects.

The reviewer is also correct that CTB injection in dLGN results in labeling of SST+PV-, SST+PV+ and SST-PV+ neurons. Please note that CTB can be taken up by fibers of passage that pass through the dLGN to other visual nuclei (e.g. LP). Therefore, it is impossible to conclude that all the “retrogradely” labeled cells in nRT project exclusively to dLGN. Our findings do not exclude that some PV nRT neurons may project to dLGN. This is now clarified in the text. Please see page 4, lines 104-113 and copied here for clarity:

“Furthermore, injections of the retrograde tracer cholera-toxin in dLGN resulted in retrograde staining of SST and PV neurons in the visual sector of the nRT (Figure 1—figure supplement 1). […]Retrograde labeling also revealed neurons that co-expressed SST and PV markers consistent with previous work (Clemente-Perez et al., 2017, Figure 1—figure supplement 2).”

4b. The authors should clarify why in Figure 1-supplemental figure 1, there does not appear to be co-localization between ChR2-EYFP and PV IHC in PV-Cre;ChR2-EYFP mice

There are multiple explanations for this:

1) The ChR2-eYFP construct we used is mainly expressed in processes (dendrites and axons) and only poorly in the soma, whereas the antibody against PV strongly labels somata.

2) The antibody does not penetrate through the depth of the brain slice so a z-stack projection fails to show the antibody labeling in many of the cells.

To address this concern, we are now showing one section of a z-stack rather than a z-stack projection of the section. We revised the Figure 1 Supplement 1 for clarity.

4c. The possibility of co-expression should be incorporated into the interpretation of the results.

Thank you for the suggestion. The co-expression is incorporated in the revised manuscript:

Results (page 4, lines 104-113):

“Retrograde labeling also revealed neurons that co-expressed SST and PV markers consistent with previous work (Clemente-Perez et al., 2017, Figure 1—figure supplement 2).” Discussion (page 10, line 315): “Certain cells expressed both markers (Figure 1—figure supplement 1) consistent with previous work (Clemente-Perez et al., 2017).”

We also edited the Discussion section to incorporate these results (page 10, lines 299-304; 316-318; 328-330).

4d. The possibility for interactions between PV and SST neurons in nRT should be incorporated into the authors' interpretation of their results

The possibility of interactions was incorporated in the previous version of the manuscript, see page 12 and 376 in the current version: “These findings are consistent with the possibility that the PV nRT neurons inhibit dLGN interneurons and/or SST nRT neurons, although such connections are yet to be established.”

4e. The reference to Campbell et al. 2020 regarding PV and SST expression in the visual sector of nRT should be eliminated as their comparison was between SST and NeuN

We apologize for incorrectly citing the reference. We eliminated the reference from the sentence.

4f. The figure legend in Figure 1C, D should be changed to SST for consistency. Consider changing SOM to SST throughout (ie. Figure 1—figure supplement 2).

Thank you for the suggestion. This is done.

5. It is difficult to interpret the effects of activation of SST-positive nRT neurons on population decoding and mutual information measures. These effects could simply be due to non-specific inhibition changes produced by activating a random subset of this population introducing an additional source of noise to the sensory signal. While the observation that most single-units recorded in LGN were inhibited may reduce this concern, it is unclear whether this is sufficient to address the concern. A more useful approach would be to compare sensory response properties of LGN neurons with and without SST activation. This could be done qualitatively by comparing receptive field properties of LGN neurons and the coefficient of variation (which was removed from the previous revision) could be used to support the claim that changes in response fidelity were present at the level of individual neurons and thus that the changes in population decoding were not a consequence of inhomogeneous. If the authors choose not to include such an analysis, this caveat should be included explicitly in the discussion.

To address this concern, we have followed the suggestion of the reviewer and now added figure 7—supplementary figure 3 which shows that optogenetic activation of SST neurons increases the variability of responses in dLGN neurons (page 9, line 267-273).

6. There remains some lack of clarity in the text. Some claims are very difficult to understand from what is written (see, for example, the case mentioned in Concern 1). The complexity of some claims also makes it difficult for the authors to draw specific conclusions in many cases. For instance, after discussing a variety of changes in oscillatory dynamics across several frequency bands during locomotion and non-locomotion produced by bi-directional optogenetic manipulations of nRT sub-populations the authors conclude that "SST nRT neurons are well positioned to exert powerful effects on the visual input to V1".

We revised the text to address this concern. Please see page 9, line 280 and copied here for clarity:

“These findings from optogenetic perturbation of SST nRT neurons on dLGN activity reveal that activating nRT neurons powerfully reduces the visual input to V1.”

Moreover, the relationship between oscillatory dynamics and sensory responses of individual neurons is still somewhat unclear. These challenges should be mitigated by a careful discussion of the relationship between changes in oscillatory dynamics, firing rates and the relevant neuronal populations in the thalamus and cortex and further clarification of the language reporting the results in a revised manuscript.

We added this information in the Introduction. Please see page 3, lines 66-71 and copied here for clarity:

“A positive relationship between gamma power and the encoding of memories as assayed by retrieval has been widely reported for many years in the hippocampus and closely related cortical areas (Wang, 2010 and references therein). […] For this reason, we focused our study on the role of specific nRT cell types in thalamocortical visual processing on measurements of both field potentials and the fidelity of encoding of visual information in the spike activity of V1 neurons.”

Reviewer #1:[…] The authors have comprehensively addressed the feedback of the first submission, and have delivered a much stronger manuscript as a result. I am particularly impressed by the new statistical approach and the additional sections in the methods. As such, I have no additional major suggestions. There are some thoughts on additional behavioral experiments, which I'm sure will come of no surprise to the authors.I have two remaining concerns about the statistical approach:1. Why not use the nested approach in Figure 6?2. There is a mention of Bonferroni corrections in the methods but the reported P-values are still * p < 0.05. Use of a Bonferroni correction means adjusting the alpha – reporting the adjusted alpha (0.05/number of comparisons) would make sense here.

Thank you for these suggestions. We use the nested approach in Figure 6 of the revised manuscript. We also clarified the adjusted alpha for multiple comparisons.

The authors might reconsider the opening of the paper, which largely focuses on gamma oscillations. Although this is indeed one of the main findings of the manuscript, it is not the only finding, and in this reviewer's perspective, this manuscript offers quite a bit by way of understanding the cell types within a previously unknown thalamic nucleus, and also in the mutual information approach. Regardless, the logical progression of the introduction is clear and I can appreciate the lab's prevailing interest in gamma oscillations and their role in visual attention.

We have revised the Introduction to address this concern (page 3, lines 66-71).

One small suggested change: the header "Optogenetic activation of SST nRT but not PV nRT neurons diminishes encoding ability of BS cells in V1" is a bit misleading, because it impacts both BS and NS cells. I suggest, "Optogenetic activation of SST nRT but not PV nRT neurons diminishes encoding ability of both BS and NS cells in V1."

Thank you for the suggestion. This is done (page 6, line 189).

Lastly, the authors have provided "source data" but this is simply a word document of the two tables in the manuscript. I do not see a spreadsheet of source data as is referenced in "Data Availability" statement on the cover sheet. The authors might consider making their data and relevant code available.

This is included.

Reviewer #2:Many of the concerns raised by the reviewers have been adequately addressed. However, there is still confusion about the exclusivity of PV and SOM neurons in nRT. It is also not clear how such heterogeneity or lack thereof, impact the results and interpretation. The authors state that the two groups are mutually exclusive but that a subgroup "show both markers". Which is it and how does this impact the interpretation of results? The bottom line based on the authors results is to acknowledge anatomically there is sparse co-expression in the visual sector, but functionally SST projecting neurons seem to regulate dLGN activity related to gamma.

We apologize for lack of clarity. We revised the text to clarify the existence of populations that express PV or SST or both markers. We also clarified that the most important result of the study is that opsin expression in *Pvalb-Cre* and *Sst-Cre* neurons of the nRT yields distinct results demonstrating that the opsin is expressed in distinct neuronal populations in *Pvalb-Cre* and *Sst-cre* mice. Notably, the fact that *Sst-cre* and *Pvalb-Cre* cells led to distinct functional effects on V1 cortex demonstrate that the opsin targets distinct cell populations despite the existence of a subgroup that expresses both markers. We clarify that the goal of our manuscript is not to quantify the percentage of cells based on their expression of certain markers, but rather to understand whether *Sst-cre* and *Pvalb-Cre* cell populations in the visual nRT have distinct functional effects on V1, similar to what has been shown for the somatosensory sector of the nRT (Clemente-Perez et al., 2017). We edited the text to clarify the goals and the novelty. Please see page 9, lines 298-304 and copied here for clarity:

“SST and PV neurons have been defined by antibody staining of the protein markers, by genetic labeling throughout development, and by the recombinase of activity in adulthood. […] The present study reveals distinct functional effects of Sst-Cre and Pvalb-Cre expressing neurons.”

Other related issues are listed below.The reference to Campbell et al. 2020 is inaccurate, since in this paper they showed about 80% of neurons in the visual sector co-express SST and NeuN (not SST and PV). There are several instances throughout the paper that mistakenly cite Campbell et al. 2020 as evidence for PV and SST co-expression. This needs to be amended.

We sincerely apologize for this mistake. We addressed this important concern in the revised manuscript.

In the supplemental figure 1, the authors showed with IHC, the labeling of PV in SST-Cre ChR2-EYFP mice. Based on the high-power images it appears that SST and PV are mutually exclusive in the visual sector of nRT (head), but it doesn't appear this was quantified; was there any evidence of co-expression (as asked by one of the reviewers) in the visual sector? CTB experiments seem to reveal some co-expression but from the images it's difficult to evaluate how prevalent.

As the reviewer noted, PV and SST neurons defined by cre recombinase expression in adulthood can to some extent express the other marker as previously published (Clemente-Perez et al., 2017) and acknowledged throughout our manuscript (see Introduction page 2, Results page 3-4, and Discussion page 10). What is novel in our manuscript is that perturbing *Pvalb-Cre* and *Sst-Cre* neurons that express ChR2 (through viral injection) has distinct effects on the V1 cortex, which suggests that the effects on V1 are mediated by distinct neuronal outputs. The significance of the findings of this manuscript concern the functional effects.

The reviewer is also correct that CTB injection in dLGN results in labeling of SST+PV-, SST+PV+ and SST-PV+ neurons. Please note that CTB can be taken up by fibers of passage that pass through the dLGN to other visual nuclei (e.g. LP). Therefore, it is impossible to conclude that all the “retrogradely” labeled cells in nRT project exclusively to dLGN. Our findings do not exclude that some PV nRT neurons may project to dLGN. This is now clarified in the text. Please see page 4, lines 104-113 and copied here for clarity:

“Furthermore, injections of the retrograde tracer cholera-toxin in dLGN resulted in retrograde staining of SST and PV neurons in the visual sector of the nRT (Figure 1—figure supplement 1). […] Retrograde labeling also revealed neurons that co-expressed SST and PV markers consistent with previous work (Clemente-Perez et al., 2017, Figure 1—figure supplement 2).”

In the supplemental figure 1, I am puzzled by the lack of co-expression between PV-Cre/ChR2 eYFP and PV IHC. Do the authors have an explanation?

There are multiple explanations for this:

1) The ChR2-eYFP construct we used is mainly expressed in processes (dendrites and axons) and only poorly in the soma, whereas the antibody against PV strongly labels somata.

2) The antibody does not penetrate through the depth of the brain slice so a z-stack projection fails to show the antibody labeling in many of the cells.

To address this concern, we are now showing one section of a z-stack rather than a z-stack projection of the section. We revised the Figure 1 Supplement 1 for clarity.

While the authors show that PV projections largely project to PV and not dLGN, they still failed to address whether intrinsic connections between PV and SOM in nRT could complicate their optogenetic manipulations.

Thank you for the suggestion. The co-expression is incorporated in the revised manuscript:

Results (page 4, lines 104-113): “Retrograde labeling also revealed neurons that co-expressed SST and PV markers consistent with previous work (Clemente-Perez et al., 2017, Figure 1—figure supplement 2).” Discussion (page 10, line 315): “Certain cells expressed both markers (Figure 1—figure supplement 1) consistent with previous work (Clemente-Perez et al., 2017).”

Figure 1 C-D, the bar graphs refer to SOM, for consistency it should be changed to SST (see also figure 2 supplement panels C-E).

Thank you. This is done.

Reviewer #3:The author's study brings together a set of interesting experiments designed to investigate the role of the inhibitory nucleus reticularis of the thalamus (nRT) in controlling sensory processing in the visual cortex and relate this control to changes in gamma oscillatory dynamics that have been tied to attentional control and state-specific sensory processing. By specifically controlling molecularly defined groups of nRT neurons using optogenetic techniques, they find that somatostatin- (SST-) positive neurons play a distinct role in regulating visual cortical oscillations as well as population encoding across multiple conditions. Although this is a valuable discovery, there are some conceptual questions that remain unanswered. First, it is unclear how to interpret the changes in population decoding observed in V1 following SOM neuron activation. Because the optogenetic approach used would randomly activate a subset of SST neurons, the change in the representation they estimate may simply represent an injection of noise into the signal arising from the non-specific inhibitory input. While their linear decoding approach suggests that the effect is not simply due to reduced spiking, the issue of non-specific inhibition is difficult to exclude based on the findings presented. In addition, while their optogenetic experiments convincingly show that activation of SST-positive nRT neurons has a much stronger effect on cortical activity compared with similar manipulation of parvalbumin- (PV-) positive nRT neurons, the anatomical tracing results they presented leave some uncertainty as to whether this population is the main inhibitory projection from the visual subdivision of the nRT to the primary visual thalamus (the lateral geniculate nucleus, LGN). In particular, without quantification the retrograde tracing shown cannot exclude the possibility that PV neurons also substantially project to the LGN.

We agree and had made this point in the previous version of the manuscript: see page 11: “These findings are consistent with the possibility that the PV nRT neurons inhibit dLGN interneurons”. In the revised manuscript, we clarified 1) the presence of neurons that express both PV and SST markers; 2) the novelty of the study is not a quantification of the percentage of distinct markers in the nRT but the finding that *Pvalb-Cre* and *Sst-Cre* neurons have distinct functions and their perturbation affects the V1 cortex differently, and this finding is robust and needs to be taken into consideration when using the different mouse *Cre* lines for studying the circuitry and function of the nRT.

The reviewer is also correct that CTB injection in dLGN results in labeling of SST+PV-, SST+PV+ and SST-PV+ neurons. Please note that CTB can be taken up by fibers of passage that pass through the dLGN to other visual nuclei (e.g. LP)., Therefore, it is impossible to conclude that all the “retrogradely” labeled cells in nRT project exclusively to dLGN. Our findings do not exclude that some PV nRT neurons may project to dLGN. This is now clarified in the text. Please see page 4, lines 104-113 and copied here for clarity:

“Furthermore, injections of the retrograde tracer cholera-toxin in dLGN resulted in retrograde staining of SST and PV neurons in the visual sector of the nRT (Figure 1—figure supplement 1). […] Retrograde labeling also revealed neurons that co-expressed SST and PV markers consistent with previous work (Clemente-Perez et al., 2017, Figure 1—figure supplement 2).”

Finally, although the authors do show that both gamma oscillations and population encoding are impacted by changes in SST activity, how these changes related to each other remains unclear.

The reviewer is correct about the current state of knowledge. We added this information in the introduction. Please see page 3 and copied here for clarity:

“A positive relationship between gamma power and the encoding of memories as assayed by retrieval has been widely reported for many years in the hippocampus and closely related cortical areas (Wang, 2010 and references therein). […] For this reason, we focused our study on the role of specific nRT cell types in thalamocortical visual processing on measurements of both field potentials and the fidelity of encoding of visual information in the spike activity of V1 neurons.”

While valuable results are presented in the manuscript some technical and conceptual issues remain that complicate the interpretation of these results. Although revisions made to the previous submission have improved the manuscript, particularly by appropriately weakening and/or clarifying some claims, some major concerns still present issues for the overall manuscript which it would be important to address. These concerns are described in detail below:1. In the previous submission concerns were raised regarding the claim that the majority of neurons projecting from the visual nRT to the LGN are SST-positive, with PV-positive neurons projecting instead to higher-order visual thalamic neurons (Previous Concern 7 and 8) and that the injection site used for the retrograde labelling may not have fully addressed these concerns (Previous concern 9). To address these concerns, the authors performed retrograde labelling experiments using cholera-toxin (CTB-Alexa488) injected into the LGN followed by histological assessment of labelled neurons to determine whether they were PV- or SST- positive. Based on these experiments, it appears that both cell types project to LGN. However, based on their earlier viral tracing experiments, they still conclude that the predominant projection is from SST-positive neurons. While the earlier results do provide qualitative support for this conclusion, it would still be important to have some quantification in order to make the claim that input is "mainly" from SST.

We agree and had made this point in the previous version of the manuscript: see page 11: “These findings are consistent with the possibility that the PV nRT neurons inhibit dLGN interneurons”. In the revised manuscript, we clarified 1) the presence of neurons that express both PV and SST markers; 2) the novelty of the study is not a quantification of the percentage of distinct markers in the nRT but the finding that Pvalb-cre and SST-cre neurons have distinct functions and their perturbation affects the V1 cortex differently, and this finding is robust and needs to be taken into consideration when using the different mouse cre lines for studying the circuitry and function of the nRT.

The reviewer is also correct that CTB injection in dLGN results in labeling of SST+PV-, SST+PV+ and SST-PV+ neurons. Please note that CTB can be taken up by fibers of passage that pass through the dLGN to other visual nuclei (e.g. LP)., Therefore, it is impossible to conclude that all the “retrogradely” labeled cells in nRT project exclusively to dLGN. Our findings do not exclude that some PV nRT neurons may project to dLGN. This is now clarified in the text. Please see page 4, lines 104-113 and copied here for clarity:

“Furthermore, injections of the retrograde tracer cholera-toxin in dLGN resulted in retrograde staining of SST and PV neurons in the visual sector of the nRT (Figure 1—figure supplement 1). […] Retrograde labeling also revealed neurons that co-expressed SST and PV markers consistent with previous work (Clemente-Perez et al., 2017, Figure 1—figure supplement 2).”

2. Previously, a concern was raised regarding the statistical significance of changes in gamma power produced by suppression of SST neurons (Previous concern 2). This remains an important claim for the overall message but, while the authors rebuttal appears to state that additional statistical analysis has clarified this point, this is not at all clear from the text. The rebuttal response refers to Figure 5A which is described in the main text as follows: "We found that inhibiting eNpHR-expressing SST nRT neurons at baseline produced a consistent, but insignificant, increase in both gamma and spiking activity of cells in V1 and a reduction in theta and beta (Figure 5A, Table 2, Figure 5—figure supplement 1, Figure 2-source data 1).…". Based on this, it is very hard to determine whether any significant change occurred (indeed, the change is described as "insignificant"). Moreover, the analysis in the figure itself appears to be comparing inhibition of PV versus SST rather than making any comparison with baseline although it is difficult to be certain due to some overall issues with labelling (see major concern 4). This key point needs substantial clarification in the main text and legend.

We have clarified the basis of statistical comparisons.

3. It is difficult to interpret the effects of activation of SST-positive nRT neurons on population decoding and mutual information measures these effects could simply be due to the fact that non-specific inhibition changes produced by activating a random subset of this population introduce an additional source of noise to the sensory signal. While the observation that most single-units recorded in LGN were inhibited may reduce this concern, it is unclear whether this is sufficient to address the concern. A more useful approach would be to compare sensory response properties of LGN neurons with and without SST activation. This could be done qualitatively by comparing receptive field properties of LGN neurons and the coefficient of variation (which was removed from the previous revision) could be used to support the claim that changes in response fidelity were present at the level of individual neurons and thus that the changes in population decoding were not a consequence of inhomogeneous inhibitory input. In any case, however, it would be worth including this caveat in the discussion.

To address this concern, we have followed the suggestion of the reviewer and now added figure 7—supplementary figure 3 which shows that optogenetic activation of SST neurons increases the variability of responses in dLGN neurons (page 9, line 267-273).

4. Despite some improvement compared to the previous submission, there is still substantial lack of clarity in the text. Some claims are very difficult to understand from what is written the text (see, for example, the case mentioned in Major Concern 2). The complexity of some claims also appears to make it difficult for the authors to draw specific conclusions in many cases. For instance, after discussing a variety of changes in oscillatory dynamics across several frequency bands during locomotion and non-locomotion produced by bi-directional optogenetic manipulations if nRT sub-populations the authors conclude that "SST nRT neurons are well position to exert powerful effects on the visual input to V1". Moreover, as before, the relationship between oscillatory dynamics and sensory responses of individual neurons is still somewhat unclear. While these challenges could have been mitigated by a careful discussion of the relationship between changes in oscillatory dynamics, firing rates and the relevant neuronal populations in the thalamus and cortex, such a discussion does not seem to have been included in the current manuscript.

This comment was addressed above.